

# Nitrogen leaching from natural ecosystems under global change: a modelling study

Maarten C. Braakhekke[1,2], Karin T. Rebel[1], Stefan C. Dekker[1], Benjamin Smith[3], Arthur H.W. Beusen[2,4], and Martin J. Wassen[1]

[1]Copernicus Institute of Sustainable Development, Faculty of Geosciences, Utrecht University, Heidelberglaan 2, 3584 CS, Utrecht, the Netherlands
[2]PBL Netherlands Environmental Assessment Agency, Postbus 30314, 2500 GH, The Hague, the Netherlands
[3]Department of Physical Geography and Ecosystem Science, Lund University, 22362, Lund, Sweden.
[4]Department of Earth Sciences, Geochemistry, Faculty of Geosciences, Utrecht University, P.O. Box 80021, 3508 TA, Utrecht, the Netherlands

*Correspondence to*: Maarten Braakhekke (maarten.braakhekke@gmail.com), Karin Rebel (k.t.rebel@uu.nl)

**Abstract.** In order to study global nitrogen (N) leaching from natural ecosystems under changing N deposition, climate, and atmospheric $CO_2$, we performed a factorial model experiment for the period 1901–2006 with the N-enabled global terrestrial ecosystem model LPJ-GUESS. In eight global simulations we used either the true transient time series of N deposition, climate, and atmospheric $CO_2$ as input, or kept combinations of these drivers constant at initial values. The results show that N deposition is globally the strongest driver of simulated N leaching, individually causing an increase of 88 % by 1997–2006, relative to pre-industrial conditions. Climate change led globally to a 31 % increase in N leaching, but the size and direction of change varied among global regions: leaching generally increased in regions with high soil organic carbon storage or high initial N status, and decreased in regions with a positive trend in vegetation productivity or decreasing precipitation. Rising atmospheric $CO_2$ generally caused decreased N leaching (33 % globally), with strongest effects in regions with high productivity and N availability. All drivers combined resulted in a rise of N leaching by 73 % with strongest increases in Europe, eastern North America and South-East Asia, where N deposition rates are highest. Decreases in N leaching were predicted for the Amazon and Northern India. We further found that N loss by fire regionally is a large term in the N budget, associated lower N leaching, particularly in semi-arid biomes. Predicted global N leaching from natural lands rose from 13.6 Tg N yr⁻¹ in 1901–1911 to 18.5 Tg N yr⁻¹ in 1997–2006, accounting for land-use changes. Ecosystem N status (quantified as the reduction of vegetation productivity due to N limitation) shows a similar positive temporal trend but large spatial variability. Interestingly this variability is more strongly related to vegetation type than N input. Similarly, the relationship between N status and (relative) N leaching is highly variable due to confounding factors such as soil water fluxes, fire occurrence, and growing season length. Nevertheless, our results suggest that regions with very high N deposition rates are approaching a state of N saturation.



# 1 Introduction

During the last century, availability of mineral nitrogen for ecosystems across the globe has risen dramatically, mainly due to production and application of fertilizers and increasing atmospheric deposition, caused by N emission from fossil fuel combustion and agriculture (Bouwman et al., 2013b; Galloway et al., 2004). This increased N availability is thought to
enhance terrestrial productivity and carbon uptake by relieving N limitation of natural ecosystems (Zaehle and Dalmonech, 2011). Excessive soil N input may, however, lead to N export, mainly as nitrate ($NO_3^-$), to ground- and surface water by leaching and lateral runoff. This results in a range of negative impacts on the environment and human health, such as eutrophication of fresh water and coastal ecosystems, fish kills, and reduction of drinking water quality (Rabalais, 2002; Schlesinger, 2009). In regions with severe ground- and surface water pollution most N export originates from agricultural
land (van Egmond et al., 2002); hence these systems have been the focus of studies quantifying N budgets (e.g. Velthof et al., 2009). From a global perspective, however, natural ecosystems are a considerable source of N input to the hydrological system (Beusen et al., 2016a; van Drecht et al., 2003). In many ecosystems the combined input from biological N fixation and atmospheric deposition now exceeds the plant and microbial demand, and in some cases rivals fertilizer application in croplands.

With increasing N input, the capacity of ecosystems to retain N decreases, resulting in larger leaching losses. However, the relationship between N inputs and mineral N leaching (hereafter simply "N leaching") is complex and non-linear, depending on factors such as vegetation type, climate, and soil properties. Insights from N manipulation experiments and measurements along N deposition gradients have spawned the concept of "N saturation", a state where N availability exceeds plant and soil microbial demand (Aber et al., 1989; Ågren and Bosatta, 1988). Since temperate forests have seen the largest increases in N
deposition, previous work on N saturation and leaching has largely focused on these ecosystems. However, N deposition is spreading to regions that were previously less affected, including boreal, tropical, and (semi-) arid ecosystems (Galloway et al., 2004; Lamarque et al., 2013a). Response of N leaching in these ecosystems is likely to differ from that in temperate forests. For example, in many tropical forests N is not a limiting nutrient, due to high rates of biological N fixation and the limited phosphorous availability (Vitousek and Howarth, 1991). These ecosystems may thus be naturally close to N
saturation (Matson et al., 2002) and N leaching is likely to be more responsive to changes in deposition (Matson et al., 1999). Grasslands, on the other hand, are usually N limited, but tend to occur in drier regions, where N losses are generally dominated by gaseous soil emissions (Bai et al., 2012) and emissions due to fire (Butterbach-Bahl et al., 2011).

Development of future N leaching is further influenced by global changes that affect terrestrial ecosystems, most importantly rising atmospheric $CO_2$ and temperature. Increasing $CO_2$ concentrations generally stimulate vegetation productivity (Norby
and Zak, 2011), leading to increased N uptake (Finzi et al., 2007) and decreased N leaching (de Graaff et al., 2006; Hagedorn et al., 2000). The effects of higher temperatures on N leaching are more ambiguous: several warming experiments found a positive effect due to increased N mineralization (Beier et al., 2008; Rustad et al., 2001); but also absence of a



response (Beier et al., 2008) or even a negative effect due to increased vegetation activity (Patil et al., 2010) has been observed.

The combined effect of (changes in) drivers as well as ecosystem and soil properties results in complex spatial and temporal patterns of N leaching rates. Global prognostic models can help understand these patterns since they provide a means of

upscaling process understanding from observational and experimental studies in order to assess N-cycling at large spatial scale. Many modelling studies on N leaching have been presented in the past decades, for both agricultural and natural ecosystems (e.g. Aber et al., 1997; Groenendijk et al., 2005; Li et al., 2006). However, the majority of these models are computationally intensive and require site specific calibration; hence they are difficult to apply at global scale. Global prognostic models have been presented by van Drecht et al. (2003) and recently by Beusen et al. (2016), which provide

spatially explicit estimates of N export to the hydrological system by partitioning N budgets into various flows, including leaching. While informative, these approaches rely on an equilibrium representation of terrestrial ecosystems and thus cannot represent changes in ecosystem N storage and N status. In this context global terrestrial ecosystem models that explicitly include N cycling represent a complementary alternative. While less suited for site level application, these models include the most important ecological processes and feedbacks that influence N leaching, and are thus useful for examining spatial

and temporal patterns and sensitivity to environmental factors. The main motivation to develop coupled C-N models has been to represent the constraint of N limitation on vegetation productivity and land C uptake (Zaehle and Dalmonech, 2011). Evaluation and application has therefore focused on variables related to C cycling (e.g. primary productivity, N use efficiency) rather than N cycling. N leaching, while sometimes reported in global modelling studies, does generally not receive specific attention (Gerber et al., 2010; Jain et al., 2009; Smith et al., 2014; Zaehle et al., 2014).

This paper presents a global modelling study into N leaching from natural ecosystems, and its response to environmental drivers. We aimed to answer the following questions: 1) what is the effect of environmental drivers, most importantly N deposition, climate, and atmospheric $CO_2$ concentration, on N leaching from natural ecosystems? And 2) what is the current N status of natural ecosystems? We used LPJ-GUESS, a dynamic vegetation/ecosystem model optimized for regional and global studies that simulates terrestrial vegetation dynamics and biogeochemical cycles. The model has recently been

extended to represent plant and soil N cycling and N limitations on plant productivity and carbon fluxes (Smith et al., 2014). The N-enabled version has been tested based on both site-level and regional observations (Smith et al., 2014; Wårlind et al., 2014) and includes the main processes underlying large scale patterns and global trends at decadal to centennial time scale of N leaching in response to drivers, which is the focus of this study. We present results from a global historical simulation, focusing on natural vegetation for the period 1901–2006. Predicted vegetation productivity and N leaching are compared to

previously published estimates from measurements and models. Furthermore, to study the individual and combined effects of the main drivers of N leaching—N deposition, climate, and atmospheric $CO_2$—we performed a full factorial experiment in which either the true transient time series were used for these drivers, or a trend-free time series, representative for pre-



industrial conditions. We discuss the effects of these factors on N leaching in the context of insights from field observations, manipulation experiments and other modelling studies.

## 2 Methods

## 2.1 LPJ-GUESS

Here a brief overview of the LPJ-GUESS model is provided, focusing on processes that are most relevant for N cycling. A complete description of the model can found in (Smith et al., 2014) and references therein as well as supplemental text S1.

### 2.1.1 General description

LPJ-GUESS (Lund-Potsdam-Jena General Ecosystem Simulator; Smith et al., 2001) simulates vegetation dynamics and biogeochemical fluxes of C and N in terrestrial ecosystems and employs generalized biome- or global-scale
parameterizations of component ecosystem processes, allowing it to be employed without recalibration globally or for any large region. LPJ-GUESS has been used extensively for studies from site to global scales. It is forced by climate variables, $CO_2$ concentration, and N deposition and runs with a daily time step, except for C allocation, vegetation dynamics, and disturbances, which are resolved annually. Our simulations focused on natural vegetation, i.e. croplands were not considered. Eleven plant functional types (PFTs) were included, representing vegetation in temperate, tropical and, boreal wooded
ecosystems and grasslands. The model predicts the occurrence of each PFT based on bioclimatic limits and competition with other PFTs for light and soil resources. Contrary to most global ecosystem models, LPJ-GUESS explicitly represents the age distribution dynamics (demography) of woody PFTs and variations in stand development across landscapes, shown to be important for carbon and nutrient balance (Haverd et al., 2014; Wolf et al., 2011). The model simulates trees of different cohorts (age classes) which are each represented by an average individual for each age class of each of a number of co-
occurring PFTs. Mortality and establishment of the individuals are implemented in a stochastic fashion, as are fire (modeled according to Thonicke et al. (2001)) and other disturbances. Sub-grid variability resulting from landscape heterogeneity and differences in disturbance history are accounted for by simulating a predefined number of replicate "patches" (area 0.1 ha) per grid cell. The conditions for all patches within a grid cell are identical but differences arise from the stochastic calculations. Within each patch LPJ-GUESS simulates fluxes of C, water and N in vegetation and soil, based on descriptions
of the key controlling processes, including photosynthesis, plant C allocation, autotrophic respiration, evapotranspiration, percolation, lateral runoff, and soil carbon cycling. The soil hydrological calculations are described in more detail in Gerten et al. (2004) and Olin et al. (2015). The simulation is initialized with a 500 year spin-up to accumulate vegetation and soil C and N pools in equilibrium with the initial forcing. During this phase the model is forced by a trend-free time series (here 10 years) of annually-varying inputs.



### 2.1.2    N cycling module

In LPJ-GUESS ecosystem N is present in vegetation biomass and in the soil in mineral and organic form. In the model version employed for our study, mineral soil N is represented by a single pool; i.e. different N species such as ammonium and nitrate, and transformation between these are not distinguished. Atmospheric N deposition is added to the soil mineral N
pool, as is biological N fixation (BNF), which is calculated as a linear function of evapotranspiration following an empirical large-scale relationship identified by Cleveland et al. (1999). Root uptake transfers N from the soil mineral N pool to vegetation on a daily time step. Plants take up N from the mineral soil pool in order to maintain optimal leaf N content required for photosynthesis (modeled according to Haxeltine and Prentice, (1996)). Following Meyerholt and Zaehle, (2015), C:N ratio of non-leaf biomass pools is fixed. If insufficient N is available the plant experiences N stress and photosynthesis
is reduced. To this end the model calculates an "N limitation factor" equal to the ratio of the true $V_{max}$ and the $V_{max}$ in absence of N limitation (both without water limitation). Here, $V_{max}$ is the carboxylation capacity of Rubisco. Additionally, different PFT cohorts compete for uptake of soil N, with grass PFTs being more competitive than tree PFTs.

N stored in vegetation is returned to the soil in organic form in conjunction with biomass turnover due to senescence, mortality, and disturbance. Litter and soil organic matter (SOM) dynamics follow the CENTURY model (Parton et al.,
1993). Gaseous N loss during nitrification and denitrification is accounted for by a 1 % reduction of the daily N mineralization. Organic N leaching occurs as a fraction of the soil microbial N pool, determined by the percolation rate and the soil sand fraction. Mineral N leaching is calculated as a fraction of the mineral N pool equal to the relative water loss by percolation and interflow. N loss due to surface runoff is not considered. Finally, fire events cause loss of vegetation N, assumed to be emitted in gaseous form.

## 2.2    Global simulations

### 2.2.1    Forcing

The model was run on a 0.5° × 0.5° global grid. Climate forcing (mean monthly fields of temperature, precipitation, cloud fraction, and number of rain days per month) was taken from the Climate Research Unit (CRU) TS 3.0 data set (Mitchell and Jones, 2005; supplemental Figure S1b-c, S2, S3), and was interpolated to daily values. Atmospheric $CO_2$ concentration was
input as global means, varying annually (supplemental Figure S1d). Spatial fields (interpolated to 0.5° × 0.5° resolution) of atmospheric N deposition were taken from the ACCMIP dataset (Lamarque et al., 2013; Figure 1, S1a), which provides annual cycles with  monthly time steps for decadal intervals. Ecosystem N input by deposition was not adjusted for leaf morphology. During the spin-up the model was forced by climate data for 1901–1910 cycled repeatedly, mean atmospheric N deposition for 1850–1860 and atmospheric $CO_2$ for 1901 (296 ppmv).





### 2.3 Factorial experiment

To disentangle the effects of N deposition, climate, and atmospheric $CO_2$ concentration on N leaching, we compared results for a series of simulations in which the model was forced either by the true, transient values for these drivers, or trend-free time series as during the spin-up. We performed a full factorial experiment for the three drivers, resulting in the eight simulations shown in Table 1. Note that for this purpose atmospheric $CO_2$ concentration is not considered a climate variable. Herein we shall refer to simulation +Ndep +clim +$CO_2$ as the "true historical simulation", and -Ndep -clim -$CO_2$ as the "control simulation". These runs were performed with 20 replicate patches (section 2.1.1); all others with 10 patches, to limit computation time.

For the analysis of the results we stratified the results by biome. Grid cells were classified into 17 biomes based on leaf area index of the PFTs and latitude according to the scheme presented in Smith et al. (2014), which is based on Hickler et al. (2006).

We assess ecosystem N limitation and saturation based on the N limitation factor (section 2.1.2). This quantity serves as an indicator of vegetation N status and ranges between zero, signifying null rubisco capacity due to N limitation, and one, signifying optimal rubisco capacity (no reduction due to N limitation).

Since the focus of this study is on natural ecosystems, we scale up LPJ-GUESS results to the globe assuming a world with potential natural vegetation. However, when comparing our results to other published estimates we correct them for the fraction of non-natural land based on the land use dataset of Hurtt et al. (2011). Models outputs are multiplied by natural land fractions (types "natural" and "barren") on a global $0.5° \times 0.5°$ grid.

### 2.4 Data used for model evaluation

To evaluate our results we compared several predicted variables to previously published estimates. First, predicted gross primary productivity (GPP) was evaluated based on a data-driven global product from the FLUXCOM dataset (Jung et al., 2017; Tramontana et al., 2016). This product was derived by training machine learning (ML) models on eddy-covariance measurements from the FLUXNET dataset using meteorological measurements and satellite data as input data, and subsequently running these models for a global spatio-temporal grid. Estimates were made with three ML algorithms and two flux-partitioning approaches (Tramontana et al., 2016), resulting in six global products. We compare our results to the mean GPP and use the spread as a measure of uncertainty. The ML models were trained for 18 landcover types (LCTs) individually which were combined using area fractions of these LCTs, which wre derived from MODIS satellite data. Since we consider natural vegetation only, we derived a modified GPP as the weighted mean over natural landcover types, where the weights are given by to the fraction of each LCT in a gridcell divided by summed fraction for all natural LCTs.



Second, N leaching predictions were compared to estimates from the IMAGE model published by Beusen et al. (2016). Briefly, Beusen et al. modelled N flows using an equilibrium approach which partitions soil N input from deposition and fixation to various losses, including surface runoff, denitrification, and leaching. The model also included natural ecosystems, but for the time frame of interest gridcells were treated as either fully natural or fully anthropogenic. Therefore, we applied the same mask for natural lands to our predictions of N leaching to improve comparability.

## 3     Results

Presentation of the results focuses on the last ten years of the simulation (1997–2006), and their comparison to the pre-industrial baseline; most graphs (except time series) show model outputs for this period.

### 3.1     True historical simulation

#### 3.1.1     Ecosystem N budget

Figure 2 depicts the ecosystem N budget for the 17 biomes and the world, including the simulated contribution of different export fluxes. Although within-biome variability is large, most biomes differ significantly from others for N input, N loss, and N net ecosystem exchange, as indicated by a Welch's t-test (supplemental tables S1–S3). Figure 2b shows that all biomes have an average positive N net ecosystem exchange (NEE) over the 1997–2006 period, i.e. they are retaining N. Highest rates of N deposition correspond to human population centers of Western Europe, Eastern USA, and South-East Asia (Figure 1a). Since these regions are generally predicted to have temperate forests (supplemental Figure S4), these biomes have the highest N deposition and therefore overall N input (Figure 2a, temperate broadleaf, deciduous and mixed forests). The same regions experienced the strongest increase in N deposition compared to pre-industrial conditions (Figure 1b). Most of the world has experienced some increase in N deposition, with a few exceptions, most notably Florida, USA, where an assumed reduction in biomass burning in the N emission data set used to derive N deposition leads to reduced N deposition (van Aardenne et al., 2001; Lamarque et al., 2013a). Near the end of the 20[th] century Indonesia (particularly Kalimantan) shows extremely high N deposition with rates higher than 100 kg N ha$^{-1}$ yr$^{-1}$ (not apparent in Figure 1 because the color axis is cut off), a result of high N emissions caused by the severe forest fires in Indonesia in 1997 and 1998 (J.-F. Lamarque, personal communication, 2016). Biological N fixation (BNF; supplemental Figure S5) is less localized than deposition. Due to the empirical relationship to evapotranspiration assumed by the model of Cleveland et al. (1999), it is predicted to occur most strongly in the tropics, and is almost absent in deserts (Figure 2a). However, in all biomes BNF is a less important N source than atmospheric deposition.

The relative contribution of N leaching, fire, and gaseous N loss to the total N loss varies strongly spatially (Figure 3). N leaching (further discussed in section 3.1.2) is important in temperate regions and the tropics, mainly due to high inputs. N



leaching also stands out in cold regions, particularly N. America, which is explained by temperature constraint on vegetation productivity. Conversely, in strongly arid regions (e.g. the Sahara) gaseous N emission dominates, due to low soil water fluxes. Finally, fire is an important N loss process in semi-arid regions, which mainly comprise grasslands. Predicted organic N leaching (supplemental Figure S6) is generally much lower than mineral N leaching, and is mostly negligible compared to

the overall N budget.

### 3.1.2   Mineral N leaching and N status

Figure 4a depicts N leaching for the true historical simulation. The regions where strongest N leaching rates occur generally correspond with regions of highest N deposition (Figure 1). The American and African tropics show moderately high leaching, because of high BNF rates. The highest leaching rates (up to 95 kg N ha$^{-1}$ yr$^{-1}$) occur in Indonesia due to the high N

deposition rates for the target period (see section 3.1.1). The spatial patterns in Figure 4a are largely mirrored by the N leaching : N input ratio (Figure 4b), indicating that with higher N inputs the relative importance of leaching increases. Exceptions are regions with significant N losses due to mineralization (e.g. Russia, Canada), and cold regions.

Figure 5 shows a global map of the N limitation factor, serving as an indicator of overall vegetation N status. It should be noted that in the model N limitation of photosynthesis is applied *before* moisture limitation (but *after* accounting for light

and $CO_2$ concentration). Therefore, it is possible that in regions where both factors are limiting additional N input would not result in higher productivity. Biome dependencies are apparent (c.f. supplemental Figure S4) with grasslands and high latitude ecosystems having generally strong N limitation while temperate forests tend to be closer to N saturation. Deserts have a wide range of N status values. The Saharan and North American deserts are predicted to be strongly N limited. At high latitudes the model also predicts very low N status for parts of the tundra and desert biomes. In most cases within-biome

variability partially explained by N input (supplemental Figure S8). However, the relationships differ strongly between biomes, and for the dry grassland and desert biomes it is virtually absent. Figure 5 further shows that N status has risen during the previous century, owing mainly to increasing N deposition.

N status has a profound effect on the relationship between N input and N leaching per biome (Figure 6). In strongly N limited biomes (tundra, grasslands, and boreal deciduous forest) considerably less N is lost by leaching than what is input.

Semi-arid biomes (tropical deciduous forests, savannahs, and arid shrublands) also leach less N than they receive. However, this is mainly related to strong N losses due to fire (c.f. Figure 3), which keeps these ecosystems continually in an aggrading state. The interaction between N status, fire, and N leaching is further illustrated by Figure 7. The graph suggests a non-linear relationship between N status and relative leaching losses. However, this apparent non-linearity is mainly caused by biomes with high fire frequency which reduces the N leaching : N input ratio. An ancillary simulation run without fire

disturbances resulted in a roughly linear relationship (supplemental Figure S11). Figure 7 further confirms the exceptional position of deserts, where N cycling is mainly determined by physical processes rather than vegetation.

### 3.1.3    Comparison with previous estimates

Figure 8 shows a comparison of gross primary productivity with estimates based on the FLUXCOM dataset (Jung et al., 2017). Although in general the spatial patterns are similar, LPJ-GUESS predicts lower productivity in the wet tropics and higher productivity in cold and dry regions at other latitudes. These mismatches largely compensate each other, resulting in similar estimates for global total GPP (125.1 ± 6.9 PgC yr$^{-1}$ (mean±SD) for FLUXCOM versus 116.8 PgC yr$^{-1}$ for LPJ-GUESS). A similar result was found by Piao et al. (2013) for the C-only version of LPJ-GUESS.

The predicted mineral N leaching rates shows good agreement with estimates of IMAGE (Beusen et al., 2016; Figure 9). The zonal mean by LPJ-GUESS shows a pronounced peak near the equator, which is not mirrored by N leaching from IMAGE results. This is likely explained by the high N deposition rates in Indonesia (section 3.1.1), which are not present in the N deposition map used as input for IMAGE (based on Dentener et al., 2006). Conversely, zonal mean N leaching is higher for IMAGE than for our results. The estimated global total N leaching near the end of the century by the two models compares similarly well (Figure S12b). However, before 1990 LPJ-GUESS shows substantially lower global rates. Again, this is likely explained by differences in N deposition, which are higher for the Beusen et al. study (Figure S12a). It appears that during the 20$^{th}$ century reduction of natural land and increases in N deposition approximately balance each other, resulting in roughly constant global total N leaching from natural ecosystems.

## 3.2    Factorial experiment

### 3.2.1    Changes in drivers during the simulation period

 N deposition, climate and atmospheric $CO_2$ all increased during the 20$^{th}$ century (supplemental Figure S1). Global total N deposition changed from 18.5 Tg N yr$^{-1}$ in 1850–1860 to more than 60 Tg N yr$^{-1}$ in 2000–2010, corresponding to a global mean N deposition of 1.4 kg N ha yr$^{-1}$ and 4.4 kg N ha$^{-1}$ yr$^{-1}$, respectively. Mean global land surface temperature rose by more than 1° C (supplemental Figure S1b) with strongest changes in North America and Asia (supplemental Figure S2). Global mean precipitation (supplemental Figure S1c) does not show a strong trend, but regionally strong increases and decreases occurred (supplemental Figure S3). Atmospheric $CO_2$ concentration rose from 296 to 381 ppmv (supplemental Figure S1d).

### 3.2.2    Effects on N leaching

Figure 10 depicts the N leaching difference relative to control for the single-factor simulations. N leaching for two-factor simulations is shown in supplemental Figure S14 (c.f. also supplemental Figure S13). For the simulation with true N deposition (Figure 10a) N leaching response generally follows the change in N deposition, showing higher rates in most places except in Florida, USA, where N deposition showed a historical decrease in the forcing data (see section 3.1.1). Climate change on the other hand has varying effects (Figure 10b). In regions with high organic carbon storage (Russia,



Canada; supplemental Figure S10), or high N availability (Europe) the response of N leaching tends to be positive, due to higher N release by mineralization. Several regions also show decreased leaching rates such as Northern India and Eastern Australia, which is mainly related to a reduction in precipitation. The effect of $CO_2$ increase (Figure 10c) is generally negative, with strongest reductions in regions with high productivity, such as the tropics. From the three single-factor runs it
is apparent that different drivers dominate in different regions for true historical simulation (Figure 10d); e.g. rising N deposition in Europe and East Canada/USA, $CO_2$ increase in the western Amazon, and climate (precipitation) change in Northern India.

Figure 11 shows the global total N leaching for the eight simulations, assuming a world with potential natural vegetation only. The strongest single driver at the global scale is N deposition change, which by itself causes an increase of N leaching
by 87 %. The overall effect of true climate is positive (31 % increase), indicating that the increased N mineralization outweighs possible uptake stimulation due to higher productivity. This is explained by the fact that the effects of climate on global GPP are relatively small and mostly negative (supplemental Figure S15, S16). In contrast, the $CO_2$ effect on global GPP is strong, resulting in a pronounced negative effect on N leaching at the global scale (-33 %). Note that, the opposing effects of rising $CO_2$ and climate on N leaching roughly balance each other, particularly in the first half of the 20th century.
Strong synergies between the drivers in relation to N leaching are not apparent from these results.

## 4    Discussion

### 4.1    Controls on N leaching

#### 4.1.1    N input

Figure 6 shows that there are large differences between biomes with regard to the relationship between N input (deposition +
BNF) and leaching. These differences are mainly related to N status: strongly N limited biomes have a lower slope, meaning that relatively less N will be leached out. Although very low N status is limited to a few biomes, in most grid cells N limitation is relevant to some extent; only ~8 % of the grid cells have an N status of 0.95 or higher. However, ecosystems do not need to be fully N saturated before significant leaching is simulated. For many biomes relative N loss by leaching starts to increase rapidly at around N status values of 0.6–0.7 (supplemental Figure S7) if water fluxes are high enough.
Furthermore, increases in N deposition occur mostly in regions that historically already had high deposition rates (Wårlind et al., 2014). Hence, at the global scale the increase in N deposition results in a strong increase in N leaching.

Several studies have reported an apparent threshold at 5–10 kg N ha$^{-1}$ yr$^{-1}$ in the relationship between N leaching and N deposition or throughfall in temperate forests. Below this deposition rate N leaching is constant at negligible levels, while above it leaching increases linearly with in deposition, although with high variability (Butterbach-Bahl et al., 2011; Dise et
al., 2009). Approximately the same threshold was found for $NO_3^-$ concentration in lakes and streams by Aber et al. (2003).



Furthermore, a recent study on the effect of N deposition on vegetation productivity concluded that photosynthetic capacity of forests reaches a plateau at approximately 8 kg N ha$^{-1}$ yr$^{-1}$ deposition (Fleischer et al., 2013). In the LPJ-GUESS predictions similar non-linear response is observed for the arctic/alpine tundra and arid shrubland/steppe biomes (Figure 6). For temperate forests, however, this behaviour is not apparent. Since the high within-biome variability may obscure the relationship, we plotted N deposition versus N leaching for the grid cells in Europe with temperate deciduous forest together with data from Level II sites of UN-ECE/EC Intensive Monitoring Programme (supplemental Figure S17; de Vries et al., 2003; Dise et al., 2009). This does not change the essential behaviour: there is in general no N input rate below which N leaching is negligible for all cells. However, the minimum N leaching (bottom of the cloud) shows an apparent threshold between 10 and 15 kg N ha$^{-1}$ yr$^{-1}$. Below this rate cells with negligible leaching occur while above it virtually all cells show significant leaching.

Globally, biological N fixation rate is predicted to be 18 Tg N yr$^{-1}$ and 32 Tg N yr$^{-1}$ with and without correction for non-natural land, respectively. The latter rate is considerably lower than observation based estimates of potential N fixation, which lie in the range of 100–290 Tg N yr$^{-1}$ (Cleveland et al., 1999), and is due to a known discrepancy between the evapotranspiration (AET) simulated by LPJ-GUESS (which falls within the range of observational estimates) and the values on which the Cleveland et al. N fixation-AET relationship is based (Smith et al., 2014). The low simulated values of BNF are particularly relevant in the tropics, where BNF provides the dominant source of N input for ecosystems.

### 4.1.2 Climate

Predicted N leaching response to climate is complicated by several issues. First, temperature and precipitation change simultaneously, since for consistency we chose to treat climate as a single factor in the experiment. Furthermore, while temperature change is globally relatively uniform and mostly positive (supplemental Figure S2), precipitation is spatially highly variable, showing both increases and decreases over the 20th century (supplemental Figure S3). Second, both variables affect various ecosystem processes that influence N leaching in opposing directions. Temperature influences both N mineralization and vegetation productivity, while precipitation stimulates both soil water fluxes and N input by BNF, which is linked to evapotranspiration in the model. Both variables influence fire probability, which is regionally very relevant.

Globally, climate change has a positive effect on N leaching (Figure 11), mainly due to an increase in net N mineralization (supplemental Figure S19), and a small and mostly negative effect on productivity (supplemental Figure S15). Particularly in regions with high soil organic carbon storage and regions that are N rich (Western Europe; Russia; Canada) climate has a strong positive effect on N leaching. Negative N leaching response occurs in regions with stimulated productivity, or reduced precipitation in combination with high N deposition (Northern India). Climate change also stimulates fire occurrence.



Similar to the LPJ-GUESS results, findings of observational studies on the effect of temperature on N leaching vary, depending on site conditions. Soil warming experiments generally show a stimulation of vegetation productivity, which is usually attributed to increased N availability caused by stimulated mineralization (Melillo et al., 2011; Rustad et al., 2001). Studies that report leaching fluxes support the LPJ-GUESS results in that N rich sites usually show a clear increase in N

leaching (Joslin and Wolfe, 1993; Lükewille and Wright, 1997; Schmidt et al., 2004), while N poor sites have a less strong response (Schmidt et al., 2004). It should be noted, however, that soil warming experiments may not be fully compatible with our results since they only account for the effect of increased N availability on vegetation productivity but not direct effects of increased temperature on plants.

Global long-term trends in N leaching do not appear to be related to precipitation in our study. At the regional scale,

however, changes in precipitation can be quite important for N leaching, as shown by the factorial experiment (section 3.2.2). Furthermore, spatial patterns of N leaching are strongly linked to soil water fluxes, which relate directly to precipitation, as illustrated by the strong relationship between runoff and N leaching on log-log scale (supplemental Figure S18). This relationship compares well with findings of Lewis et al., (1999).

### 4.1.3   Atmospheric $CO_2$ concentration

LPJ-GUESS predicts a considerable negative response of N leaching to changes in atmospheric $CO_2$ concentration (Figure 10c and Figure 11). Strong reductions occur in regions which have high N availability, either due to N deposition (temperate biomes) or N fixation (tropics). Furthermore, $CO_2$ also stimulates litter production, which tends to increase fire occurrence, resulting in volatilisation N losses which lead to a compensatory reduction in leaching in semi-arid biomes. Particularly in the tropics LPJ-GUESS predicts a strong GPP response to $CO_2$ increase (supplemental Figs. S15, S16). Due to the lack of

$CO_2$ enrichment experiments in the tropics, little is known about the $CO_2$ response of tropical forests (Hickler et al., 2008); hence it is unclear to what is extent this behaviour is realistic. It is likely that in reality phosphorus availability limits GPP response (Wang et al., 2010), which would presumably reduce the negative effects on N leaching.

Although there is a large body of literature on the effects of $CO_2$ enrichment on ecosystems, few studies report N leaching. Several studies found reductions in N leaching under elevated $CO_2$ (Hagedorn et al., 2000; Hungate et al., 1999; Johnson et

al., 2004) but absence of response has been reported as well (Larsen et al., 2011). In addition to enhancing plant productivity, elevated $CO_2$ tends to increase water use efficiency (Dekker et al., 2016), resulting in a reduction in transpiration. This stimulates runoff and water leaching (Betts et al., 2007), which could be expected to affect N leaching.

### 4.1.4   Fire

LPJ-GUESS predicts that fire plays an important role in the N budget of natural ecosystems (Figure 2, Figure 3;

supplemental Figure S9). Globally, fires account for 33 % of the total N loss in the period 1997–2006, approximately the



same as the contribution of N leaching. Ecosystems with frequent fires are more N limited and leach less N in the model predictions. This agrees with field studies that have found that fires lead to decreased N leaching on longer time scales (>3 years) (Johnson et al., 2007). Prescribed fires have been proposed as a measure to improve surface water quality (Fenn et al., 1998; Johnson et al., 2008). On shorter time scales, however, fire may enhance leaching due to mineralization of fire-

induced litter fall (Alexis et al., 2007; Johnson et al., 2008). Furthermore, at burned sites original vegetation is often replaced by N fixing plants, which quickly replenish lost N (Johnson et al., 2008). Neither of these processes is represented in LPJ-GUESS.

If we adjust the LPJ-GUESS prediction of global N fire loss for the fraction of non-natural land we find an estimate of 16.7 Tg N yr$^{-1}$. This compares well with observation based estimates. For example, according to the Global Fire Emissions

Database (GFED4; van der Werf et al., 2010), 16.4 Tg N yr$^{-1}$ was emitted from natural fires globally ($N_2O + NH_3 + NO_x$) in 1997–2006. Schultz et al. (2008) reported a somewhat higher value for the 1990s: 24 Tg N yr$^{-1}$ ($NH_3 + NO_x$). Although uncertainty regarding N losses by fire is large (Gruber and Galloway, 2008), the general agreement with observations strengthens our finding of the important role of fire in the terrestrial N budget. Specifically for savannah ecosystems, the influence of fire on N dynamics is further supported by field and remote sensing studies (Veldhuis et al., 2016; Chen et al.,

15 2010).

It should be noted that part of the emitted N from fire is quickly returned to the land surface by atmospheric deposition. This is not accounted for in our study, since we prescribed N deposition, based on the dataset of Lamarque et al. (2013), which is derived from simulations with an ensemble of atmospheric chemistry models. The N emission sources used to drive these models include fires (Lamarque et al., 2010) but it is unknown whether these agree with the fires predicted by LPJ-GUESS

in terms of pattern and magnitude.

## 4.2    Ecosystem N status

We have used the N limitation factor, which reflects the reduction in photosynthetic capacity due to N limitation, as a proxy for N status (Figure 5). While this quantity is in general correlated to N input, its spatial variability appears to be more strongly related to ecosystem type (supplemental Figure S8). The savannah, grassland, boreal forest, and tundra biomes show

strong limitation, while for example tropical rainforests are generally closer to saturation. This agrees well with observational findings on the prevalence of N limitation in natural ecosystems (Vitousek and Howarth, 1991). Apart from differences in climate zone, the relationship between biome and N status is the result of the interaction of several PFT specific interactions. Grass PFTs have higher productivity, resulting in a higher N demand and will thus more quickly experience N limitation. In LPJ-GUESS tundra vegetation is represented by C3 grass, which explains the strong N limitation

for this biome. Grass PFTs also experience more fire, which exacerbates N limitation in semi-arid regions. Phenology plays



a role as well—regions with a short growing season due to seasonally low temperatures or drought are less efficient in retaining N.

For similar reasons the relationship between N status and N leaching is also confounded. Although in general regions with high N status tend to lose more N by leaching according to our simulations (supplemental Figure S7), variability is large, due to the interplay of natural and ecological processes. For example, several arid regions (e.g. Chinese deserts) have generally high N status but leaching rates are quite low due to low water fluxes. On the other hand, in N limited ecosystems significant leaching can still occur (Figure 6), demonstrating that there are limits to the efficiency of plants to retain N. To some extent these results contradict the conceptual model of N saturation proposed by Aber et al. (1998, 1989) which states that N leaching occurs at significant rates only when an ecosystem is fully N saturated. This view has previously been criticized by Lovett and Goodale (2011) who pointed to studies that found that N limited ecosystems can in fact leach considerable amounts of N.

Thus, N limitation of vegetation as a proxy for N status is of limited value for predicting N leaching in a global setting. Conversely, the ratio of N leaching : N input may be a useful complementary indicator for N status. Ignoring high latitude regions, this quantity is relatively high in regions with high N inputs (e.g. Western Europe, North East USA, South East Asia, and the tropics; Figure 4b), which agrees with observational findings that the proportion of N lost by leaching increases with N deposition (Aber et al., 2003; Fenn et al., 1998).

### 4.3    Comparison with other large scale studies

As discussed in section 3.1.3, there is good spatial agreement with N leaching rates of Beusen et al. despite the differences between the models. This agreement likely results at least partially from similarities in the N deposition maps used as input (notwithstanding Indonesia, see section 3.1.1). Although different data sources were used (Lamarque et al., 2013 for our study; Dentener et al., 2006 for Beusen et al.), there is considerable agreement between the two. Conversely, earlier in the 20[th] N deposition input match less well, as does the predicted N leaching (supplemental Figure S12). This supports our finding that N input, and specifically N deposition, is in general the dominant factor driving spatial and temporal differences in N leaching rate. Nevertheless, given the high uncertainty associated with these estimates, the good agreement is encouraging.

After correction for non-natural land cover, based on the land-use dataset of Hurtt et al., (2011), we estimate a global N leaching rate of 18.5 Tg N yr$^{-1}$ from natural ecosystems for 1997–2006. Although a number of studies have quantified terrestrial N export, most of these do not distinguish contributions from different land cover types. The studies that did report the contribution of natural ecosystems found rates of a magnitude comparable to our estimate. For example, Zaehle et al. (2010) found 27 Tg N yr$^{-1}$ for the 1990s based on a simulation with the OCN land surface model. Recently, Nevison et al. (2016) reported N export (leaching + surface runoff) for 1995–2005, from a simulation with the Community Land Model





with natural vegetation only. As a rough correction for non-natural land, we may multiply their estimate (10.6 Tg N yr⁻¹) by 0.64, the ratio of our rate and the predicted N leaching for a completely natural world (28.6 Tg N yr⁻¹). This yields a value of 6.8 Tg N yr⁻¹, which is considerably lower than other published rates, including ours. The authors acknowledged that this is likely an underestimation caused by too high denitrification rates.

## 4.4 Model limitations

The LPJ-GUESS model is part of the state of the art of global ecosystem modelling and compares favourably with other models in model-data comparison studies (Zaehle et al., 2014). Nevertheless, uncertainties in modelling N cycling at the global scale are in general high. Competing model representations yield divergent results (Zaehle and Dalmonech, 2011), and available observations are currently not sufficient to identify best parametrizations. We will discuss several specific model limitations of LPJ-GUESS (also present in other models) that are particularly relevant for N leaching.

An important source of uncertainty stems from the representation of biological N fixation. As mentioned previously, BNF predicted by LPJ-GUESS is 3–9 fold lower than observation based estimates (Cleveland et al., 1999), which may be expected to cause underestimation of leaching. Furthermore, in LPJ-GUESS BNF occurs passively without explicit link to vegetation productivity or ecosystem N status. In reality plants are believed to exert some control on BNF rates based on the balance between N demand and availability (Vitousek et al., 2002). Since there is a considerable energy cost involved, plants tend to downregulate BNF under N rich conditions in favour of other pathways such as mycorrhizal, or passive, uptake (Houlton et al., 2015). Similarly, N resorption from leaves before senescence—assumed a fixed fraction of 50 % in LPJ-GUESS—tends to be lower under N rich conditions. More mechanistic descriptions of plant N acquisition have been proposed (Brzostek et al., 2014), which account for these feedbacks, and could potentially result in improved prediction of ecosystem N cycling and leaching under varying conditions.

In the current version of LPJ-GUESS all mineral nitrogen forms are lumped into a single pool. In reality inorganic nitrogen in soils exists in a range of chemical forms, most importantly nitrate ($NO_3^-$) and ammonium ($NH_4^+$). The behavior of these two species in soil differs considerably (Butterbach-Bahl et al., 2011). Ammonium is much less susceptible to leaching since soils generally have a much higher capacity for retaining cations than anions. As a result inorganic N losses occur mainly in the form of nitrate and its formation by nitrification is an important control for leaching losses (Zhang et al., 2016). Since plants and microbes preferentially take up ammonium over nitrate, nitrification rate is usually low in N limited ecosystems. Hence, it is possible that leaching losses are overestimated in N limited ecosystems, which may explain the mismatch with observations (supplemental Figure S17) at low N deposition rates. Transformation of ammonium to nitrate by nitrification, and vice versa by denitrification is associated with gaseous losses. The rates of these processes are highly dependent on substrate concentrations and soil aeration status. Particularly denitrification can be a considerable loss term in the N budget under anaerobic conditions (Bouwman et al., 2013a). Since LPJ-GUESS models gaseous losses as a fixed fraction (1 %) of



gross N mineralization, it is likely to overestimate leaching losses in soils with high N availability under wet conditions. Currently, LPJ-GUESS is being updated with an improved description of soil N cycling based on the approach of Xu-Ri and Prentice (2008) that accounts for different mineral N species and transformations between them.

Finally, LPJ-GUESS predicts that dissolved organic N (DON) leaching is generally negligible compared to the overall N
budget (supplemental Figure S6; globally 0.18 % of total N loss for 1997–2006) which likely represents an underestimation. Observations based on river concentrations indicate that DON loss can be a significant component of ecosystem N export, particularly in N limited regions (Perakis and Hedin, 2002). In process-based modelling studies DON leaching has been largely ignored as a significant N sink, which is increasingly recognized as a limitation (Nevison et al., 2016).

## 5    Conclusions

The factorial experiment with LPJ-GUESS allows us to disentangle the effects of changes in N deposition, climate, and atmospheric $CO_2$ concentration on N leaching. From a global perspective N deposition is the most important control of N leaching in our model simulations. Rising N deposition during 20[th] century has caused large increases in N leaching in many regions in the world. Rising atmospheric $CO_2$ and climate change, of secondary but not negligible importance, have a negative and a positive effect on N leaching, respectively. Although temporal trends are clear at the global scale, there are
large regional differences, even when individual drivers are considered. This variability results largely from heterogeneity of climate and N deposition changes and biome type, and causes complex spatial patterns when all three drivers are combined. These patterns would have been difficult to understand based on the true historical simulation alone. Ecosystem N status is more difficult to assess based on our results. Spatial patterns of N limitation on vegetation productivity are more strongly related to vegetation type than N input. Nevertheless, at the global scale N limitation is clearly decreasing and regions with
highest N deposition are approaching N saturation.

**Acknowledgments**

We are indebted to Mats Lindeskog, Stefan Olin, Michael Mischurow, and David Wårlind for help with setting up and running LPJ-GUESS and interpreting the results. This work was supported by funding from the global water cycle modelling project of Utrecht University and sponsored by NWO Physical Sciences for the use of supercomputer facilities. This study
contributes to the Strategic Research Areas BECC and MERGE of the Swedish Research Council. The model outputs presented in this paper will be made available for download on a public server after publication.





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



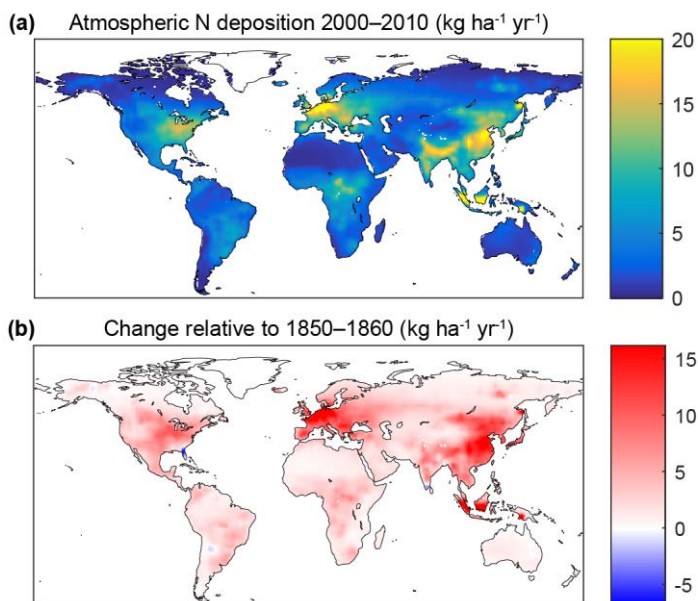

**Figure 1. Atmospheric N deposition for 2000–2010 (the last decadal interval of the ACCMIP historic N deposition data set (Lamarque et al., 2013). (a) Absolute rates. (b) Change relative to 1850–1860. For both figures the colour axis is cut off at approximately the 99 % quantile to improve readability.**





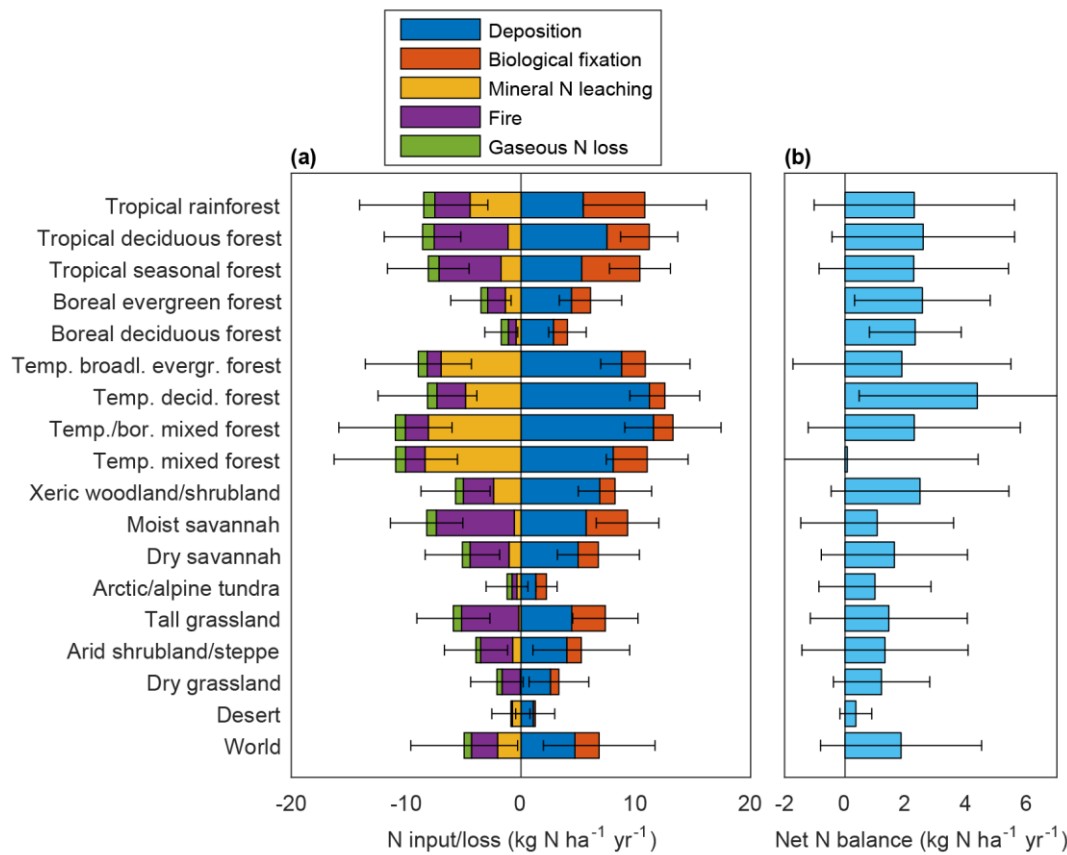

**Figure 2. Ecosystem N budget per biome for the true historical simulation (+Ndep +clim +CO$_2$) averaged over the period 1997–2006. (a) Mean N input and loss fluxes; (b) Mean net ecosystem exchange of N. Organic N leaching (not shown) is negligible for all biomes. Error bars indicate one standard deviation among area-weighted grid cell averages for the target time period.**





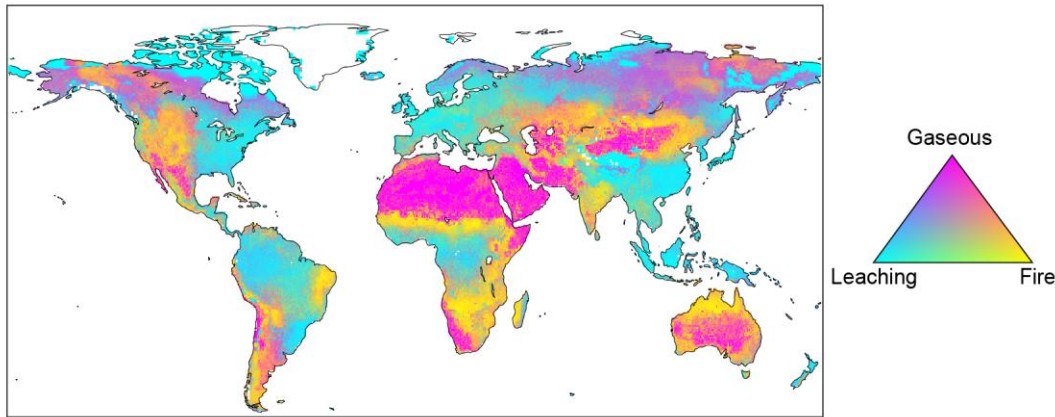

**Figure 3. Relative contribution of leaching, gaseous loss, and fire to overall ecosystem N loss for the true historical simulation (+Ndep +clim +CO2) averaged over the period 1997–2006. The colours at the three corners of the triangle indicate 100 % N loss by the corresponding process.**





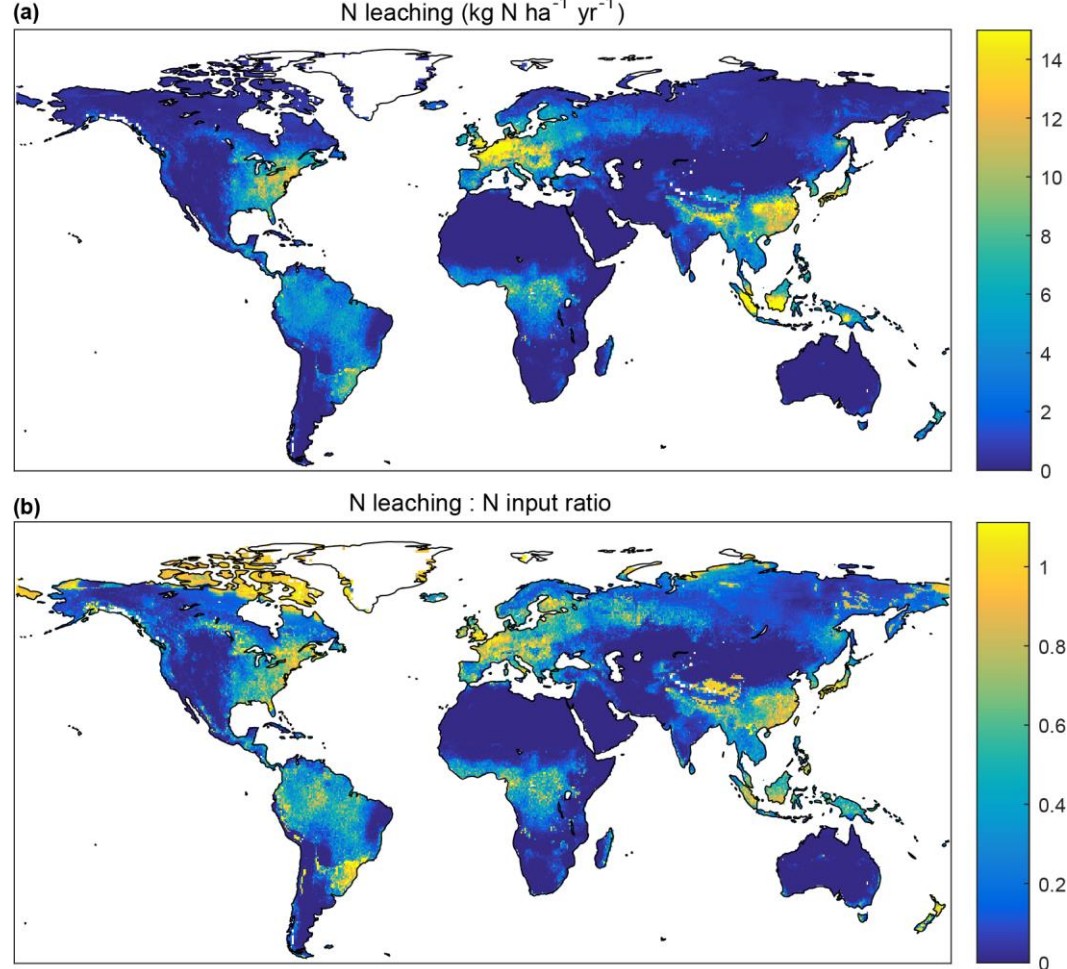

**Figure 4. Mineral N leaching (kg N ha⁻¹ yr⁻¹) for the true historical simulation (+Ndep +clim +CO2) averaged over the period 1997–2006. (a) N leaching; (b) ratio of N leaching and N input (N deposition + BNF). For readability the colour axes have been cut off at the 99 % quantile.**





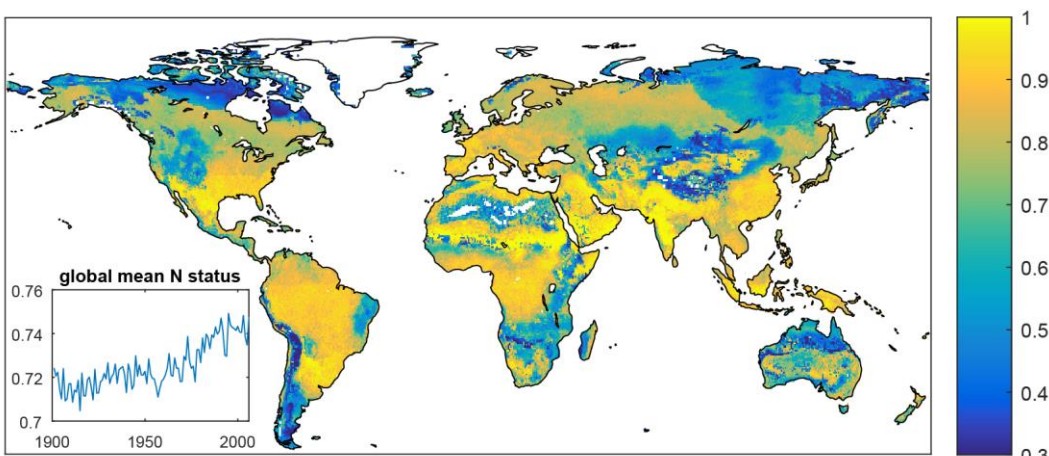

**Figure 5. Ecosystem N status for the true historical simulation (+Ndep +clim +CO$_2$) averaged over the period 1997–2006. Lower values indicate stronger N limitation (see main text). The inset shows global mean N status over time during the simulation period.**





**Figure 6. Mineral N leaching versus total N input (deposition + BNF) for the true historical simulation (+Ndep +clim +CO2) averaged over the period 1997–2006. Each point represents one grid cell. Colors indicate the mean N status. Dashed lines indicate the 1:1 relationship.**




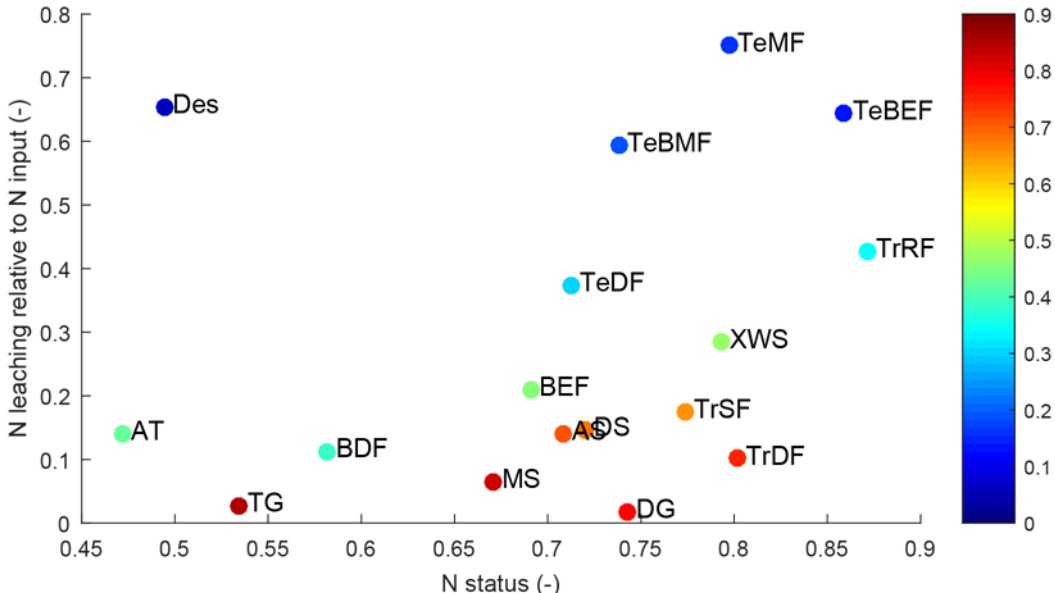

**Figure 7. N leaching relative to total N input versus N status for the 17 biomes for the true historical simulation (+Ndep +clim +CO2). Colours indicate the mean fraction of N lost by fire. TrRF: Tropical rainforest, TrDF: Tropical deciduous forest, TrSF: Tropical seasonal forest, BEF: Boreal evergreen forest/woodland, BDF: Boreal deciduous forest/woodland, TeBEF: Temperate broadleaved evergreen forest, TeDF: Temperate deciduous forest, TeBMF: Temperate/boreal mixed forest, TeMF: Temperate mixed forest, XWS: Xeric woodland/shrubland, MS: Moist savannah, DS: Dry savannah, AT: Arctic/alpine tundra, TG: Tall grassland, AS: Arid shrubland/steppe, DG: Dry grassland, Des: Desert**



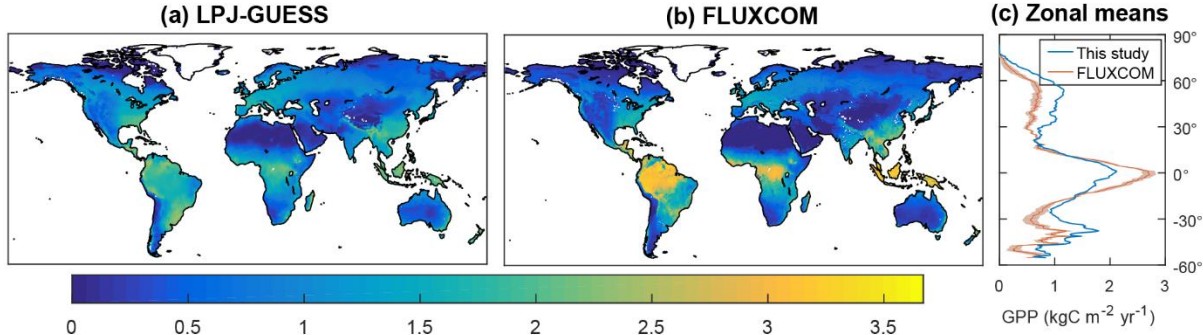

**Figure 8. Mean gross primary productivity (GPP) for 1997–2006. (a) LPJ-GUESS GPP for the true historical simulation (+Ndep +clim +CO2). (b) FLUXCOM GPP (Jung et al., 2017), mean over six approaches. (c) latitudinal averages. The shaded area for FLUXCOM indicates the 95% confidence range over the six approaches.**



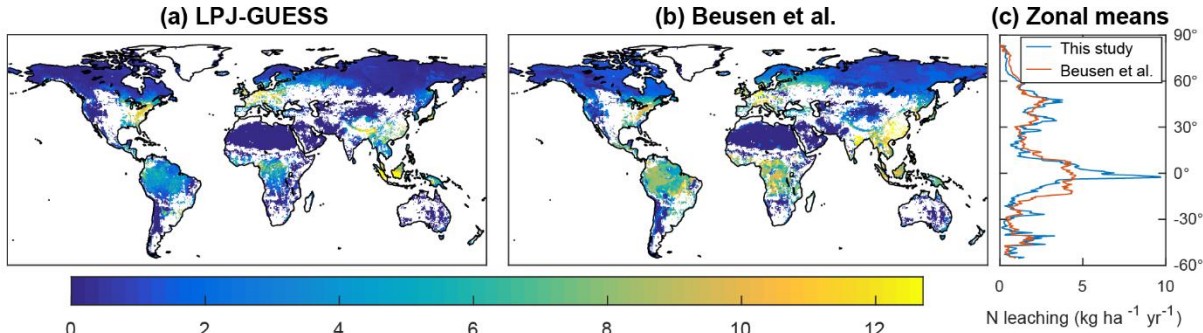

**Figure 9. Mean mineral N leaching rate for 1999–2000 compared to estimates of (Beusen et al., 2016b). (a) prediction by LPJ-GUESS for the true historical simulation (+Ndep +clim +CO2). (b) estimate of Beusen et al. for natural ecosystems only. (c) latitudinal averages. The many missing values in the maps are caused by mask for natural land cover of Beusen et al. which was applied to both datasets (see section 2.4).**





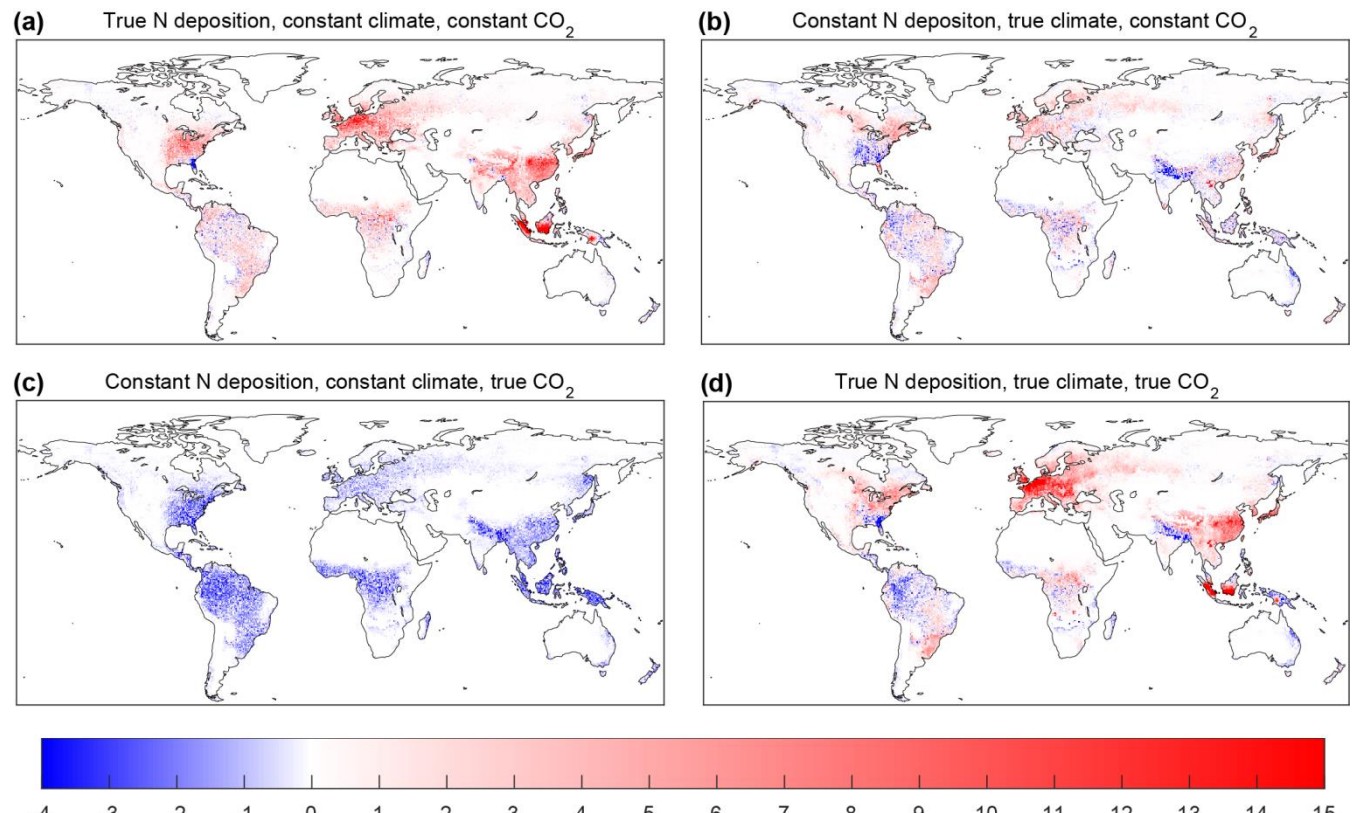

**Figure 10. Mineral N leaching difference relative to the control simulation (-Ndep -clim -$CO_2$) for the single factor runs and the true historical simulation. For readability the color axis has been cut off at the 1 % and 99 % quantile over all graphs.**



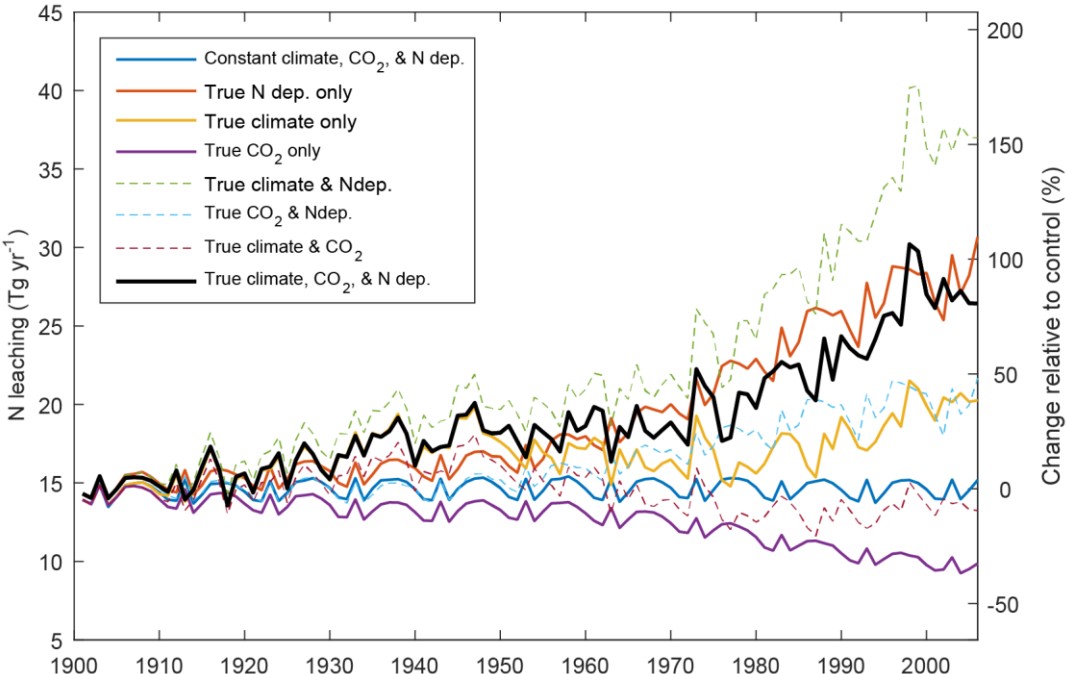

**Figure 11. Global total N leaching for the eight simulations, for a world with potential natural vegetation only. The right hand side axis depicts change relative to the mean rate for the control simulation (-Ndep -clim -CO₂).**



**Table 1. Simulation runs in the factorial simulation experiment. Note that climate comprises four variables (section 2.2.1).**

| Label | N deposition | Climate | Atmospheric $CO_2$ |
| --- | --- | --- | --- |
| **-Ndep -clim -CO₂** | constant | constant | constant |
| **+Ndep -clim -CO₂** | true | constant | constant |
| **-Ndep +clim -CO₂** | constant | true | constant |
| **-Ndep -clim +CO₂** | constant | constant | true |
| **+Ndep +clim -CO₂** | true | true | constant |
| **+Ndep -clim +CO₂** | true | constant | true |
| **-Ndep +clim +CO₂** | constant | true | true |
| **+Ndep +clim +CO₂** | true | true | true |