# Peer review of "Nitrogen leaching from natural ecosystems under global change: a modelling study"

_Earth System Dynamics, 2017_

## Referee Comment (RC1) · Anonymous Referee #1 · 16 Apr 2017

Braakhekke et al., use an Earth System Model to constrain the drivers of N leaching from 1901 to 2006. The manuscript is well written, interesting and contributes to a better understanding of the control of N leaching. Please find below a few comments that should be addressed before publication.

1) Atmospheric N deposition: theory and forcing 1.1. I am not really familiar with atmospheric N deposition, but from reading the manuscript it seems like the authors refer to nitrous oxide deposition, since N deposition is the highest in the most populated/polluted areas. I think a line or two properly defining atmospheric N deposition and its control could help better understand the paper. 1.2. It is stated that the forcing for atmospheric N deposition is taken for years 1850-1860, whereas the other forcings (climate and CO2) are taken from year 1901. I guess there is not much difference between 1860 and 1901 for N deposition or at least much less than between 1901-2006,

but a) where do estimates of N deposition for 1860 come from? And b) why would you take 1860 instead of 1901? 1.3. Following on these 2 comments, I would suggest to restructure a bit section 3.1.1. as follow: The authors could start by describing Figure 1 and explaining the origin/controls of N deposition (natural vs anthropogenic effects, then describe Figure 2 and differences for each biome.

2) Comparison with previous estimates (Section 3.1.3) Beusen estimates seem to be significantly higher than LPJ-GUESS in Equatorial regions and southern tropics. The sentence L.13-15 is very unclear to me. L. 4: "higher productivity in cold and dry regions at other latitudes." To me it seems more like in the "mid latitudes". East Australia and South Brazil/Argentina are not really cold-dry regions. I am not really convinced by this section 3.1.3. I understand the authors want to try and compare their estimates with previous studies, but here a very rough comparison is made without going really into the reasons for these differences.

3) Climate section Due to the different impacts of temperature and precipitation changes on terrestrial productivity and soil processes, this section is a bit more difficult to follow. I would suggest the authors try to use terms which indicate the direction of the change: e.g. warmer conditions increase N mineralization...

Minor comments: P2, L. 15: "with increasing N input, the capacity of ecosystems to retain N decreases.." that statement surprises me. I can understand that with increasing N input, leaching increases, but the capacity to retain N does not necessarily changes.

P4, L6. Add "be" between can and found. P6, L. 27, "we" P6, L.29: remove "to" between by and "the fraction"

Notation of "N leaching : N input ration", I don't; really like that notation as ":" also denotes a ratio. I would suggest to modify to "N leaching/N input" or "N leaching to N input ratio". P8, L. 20: add "is" after variability

Section 3.2.1, L. 18: N deposition and pCO2 increased but not climate, please modify.
**P11, L.15, year of citation for Cleveland et al., is missing**

Figures: not sure that showing the 99% quantile is the best way to go as areas with very large changes do not come out. You could also use a nonlinear color scale. Figure 8 and 9: I find it a bit confusing that panel c does not go all the way to 90S. It would be better visually to have the latitudes of the 3 plots match: i.e. if c stops at 60S, then c does not have to be shown over the whole vertical plot and would stop at 60S in panels a and b).

---

## Referee Comment (RC2) · Anonymous Referee #2 · 25 Apr 2017

Braakhekke et al. present a model study quantifying the respective contributions of changes in nitrogen (N) deposition, climate and atmospheric CO2 concentration on changes in N leaching from natural ecosystems. They find that increasing N deposition is the major driver behind simulated changes in N leaching, with smaller contribution from climate change and increasing CO2. They further highlight the role of fire in shaping N losses. The conclusions drawn have a sound basis on the the results discussed here. Overall, the manuscript is well written and clearly structured.

However, I find that the discussion of gaseous losses in the manuscript is lacking. Leaching losses are often the major loss, but gaseous losses are not negligible and can regionally dominated total N losses (Houlton et al. 2015). Accordingly, the role of gaseous losses needs to be considered when the ratio of leaching:inputs is discussed.

The author state in the introduction that "N leaching, while sometimes reported in global modelling studies, does generally not receive specific attention [...]". Therefore, due to the lack of evaluation of simulated loss terms, the reliability of global models in respect to the loss terms has to be considered low. This is true for LPJ as previous studies did not evaluate the loss fluxes sufficiently (Smith et al 2014 Warling et al., 2014). Unfortunately, the study by Braakhekke et al. does not improve this situation although data sets exist to evaluate. For example, the simulated gaseous loss fraction can be compared to reconstructions from delta 15 N measurements and models by Houlton et al. (2015) and, more recently Goll et al. (2017). There are regional differences in the dominant loss pathway between this study (Figure 3) and the mentioned studies which should be discussed.

The role of fire in shaping N loss pathways on global is a novel aspect of this study. The analysis would benefit from information on how simulated fire emissions and the contribution of wildfires to N deposition (forcing) compare to each other. Such information is completely lacking in the manuscript.

In the abstract is stated "Predicted global N leaching from natural lands rose from 13.6 Tg N yr-1 in 1901–1911 to 18.5 Tg N yr-1 in 1997–2006, accounting for land-use changes." (P1L25/26). Did the authors account for land-use change? The information in the manuscript is insufficient to tell to what extent land use (change) and for example associated nitrogen fertilization was accounted for.

Minor P2L14: reference missing P4L27: what is the criteria applied to define when the equilibrium state is reached? P5L12: how are the grass PFTs being more competitive than trees in the model? P11:13: BNF estimates were revised down since Cleveland et al. 1999. Please account for newer estimates here; for example see Vitousek et al. 2013 , Sullivan et al. 2014. P14L23: The authors state that N deposition is the dominant factor driving spatial differences in the leaching rate. This needs to shown, as this is not apparent. I rather would suspect differences in the hydrological cycle to dominate spatial leaching patterns. P14L26: how is the correction done. This should be stated

in the method sections. Figure 4: the ratios, denitrification:inputs and fire:inputs, would be interesting to see and to better understand the lack of non-linearity in the simulated leaching:input ratio (Figure 6) P14L30: the substantial underestimation of BNF in LPJ should lead to lower leaching rates. This should be discussed.

Reference: Houlton, Benjamin Z., Alison R. Marklein, and Edith Bai. "Representation of nitrogen in climate change forecasts." Nature Climate Change 5.5 (2015): 398-401. Goll, D. S., Winkler, A. J., Raddatz, T., Dong, N., Prentice, I. C., Ciais, P., and Brovkin, V.: Carbon-nitrogen interactions in idealized simulations with JSBACH (version 3.10), Geosci. Model Dev. Discuss., doi:10.5194/gmd-2016-304, in review, 2017.

---

## Referee Comment (RC3) · Anonymous Referee #3 · 5 May 2017

This paper presents a modeling study on global nitrogen leaching from natural ecosystems with an ecosystem model LPJ-GUESS. Overall, the paper is well-written and the results are informative. My major concern is on the dependence of results on model representation of nitrogen and carbon cycling process and the inherent model assumptions. Please find my specific comments below.

1. Authors conclude that atmospheric N deposition is the major driver behind nitrogen leaching globally. This is not surprising as atmospheric nitrogen deposition is the dominant N input and linearly linked to soil nitrogen storage in the model. I would suggest add more details on the mathematical formulations in representing atmospheric nitrogen deposition and nitrogen mineralization in the model. Are the results sensitive to the specific formulation of atmospheric nitrogen deposition and mineralization? Discussion on the nitrogen deposition dataset should also be added.

[Figure]

2. Table 1 lists the numerical experiments conducted in this study. I would suggest add some statements on the purpose of these experiment designs in the methodology section 2.3. For example, which combination of experiments is used to disentangle the effects of a specific environmental driver (e.g. N deposition)?

3. Increase in nitrogen deposition may potentially lead to increased plant carbon uptake and plant productivity, which would feedback to the nitrogen budget and affect leaching process. Is the impact of carbon–nitrogen interactions on N leaching process considered in this study? This aspect of carbon–nitrogen dynamics on N leaching process should at least be discussed. Please also add a table specifying carbon–nitrogen ratios for all natural plant types considered. It would be interesting to examine/discuss how plant growth regulate the simulated N leaching for different PFTs in the model.

4. The effects of fire and gaseous loss on N leaching is analyzed in the results section. But the descriptions on the representation of fire in the model is missing. In addition, it seems that the proposed numerical experiments in Table 1 doesn't consider fire?

5. In section 2.2.1, the CRU monthly climate is interpolated to daily values as inputs for the model. More details on this temporal disaggregation are required. Discussions are also needed as the simulated sensitivity of N leaching to precipitation may depend on the daily sequence of precipitation and intensity.

6. I would suggest clarify which specific aspect of N leaching is the focus of this study, the mean value or its temporal variation?

7. How is N status quantified?

8. In section 3.2.1, the statement "N deposition, climate and atmospheric CO2 all increased during the 20th century" is confusing as "climate" is a broad concept.

9. I would suggest using the percentage change (%) as the unit in Figure 10

10. The name of the model used in this study can be added in the title.

---

## Author Comment (AC1) · 17 Jul 2017

The reply to the comments is given in the attached PDF document.

Please also note the supplement to this comment:
https://www.earth-syst-dynam-discuss.net/esd-2017-7/esd-2017-7-AC1-supplement.pdf
* * *

---

## Author Response (AR1)

**Response to comments of reviewer 1**

We thank the reviewer for the praise and the many helpful suggestions, which have largely been followed. Below our reply to the specific comments, set in red font color.

Braakhekke et al., use an Earth System Model to constrain the drivers of N leaching from 1901 to 2006. The manuscript is well written, interesting and contributes to a better understanding of the control of N leaching. Please find below a few comments that should be addressed before publication.

1) Atmospheric N deposition: theory and forcing

1.1. I am not really familiar with atmospheric N deposition, but from reading the manuscript it seems like the authors refer to nitrous oxide deposition, since N deposition is the highest in the most populated/polluted areas. I think a line or two properly defining atmospheric N deposition and its control could help better understand the paper.

Section 2.1.2 has been expanded with an explanation of N deposition and biological N fixation:

1.2. It is stated that the forcing for atmospheric N deposition is taken for years 1850-1860, whereas the other forcings (climate and CO2) are taken from year 1901. I guess there is not much difference between 1860 and 1901 for N deposition or at least much less than between 1901-2006, but a) where do estimates of N deposition for 1860 come from? And b) why would you take 1860 instead of 1901?

a) The N deposition data for the complete simulation period, including the spin-up, was obtained from the ACCMIP dataset Lamarque et al. (2013).

b) The aim of the spin-up is to obtain the ecosystem state under "pre-industrial" conditions, i.e. as little influence by humans as possible. For climate and $CO_2$ concentration anthropogenic effects are relatively small before 1900, compared to 1900-2000, but for N deposition regionally quite strong changes occurred before 1900 due to land-use change (biomass burning). Hence, we chose to use the first time period in the ACCMIP dataset, 1850–1860, both for the spin-up and the simulations where N deposition was held constant.

Section 2.2.1 has been modified to clarify the above two points.

1.3. Following on these 2 comments, I would suggest to restructure a bit section 3.1.1. as follow: The authors could start by describing Figure 1 and explaining the origin/controls of N deposition (natural vs anthropogenic effects, then describe Figure 2 and differences for each biome.

Done.

2) Comparison with previous estimates (Section 3.1.3) Beusen estimates seem to be significantly higher than LPJ-GUESS in Equatorial regions and southern tropics. The sentence L.13-15 is very unclear to me.

The relevant paragraph has been rewritten and is hopefully now more clear.

L. 4: "higher productivity in cold and dry regions at other latitudes." To me it seems more like in the "mid latitudes". East Australia and South Brazil/Argentina are not really cold-dry regions. I am not really

convinced by this section 3.1.3. I understand the authors want to try and compare their estimates with previous studies, but here a very rough comparison is made without going really into the reasons for these differences.

The line the reviewer referred to has been modified according to the reviewer's suggestion. An in-depth discussion on the reason of the mismatch for GPP is difficult without a detailed analysis including site observations and other data, which is outside of the scope of this study. Presumably, re-calibration would be needed to improve the fit. The aim of the comparisons with other data sets is to see whether LPJ-GUESS predictions are sufficiently realistic for the purpose of this is study, which—in our view—is the case, despite mismatch for GPP.

3) Climate section Due to the different impacts of temperature and precipitation changes on terrestrial productivity and soil processes, this section is a bit more difficult to follow. I would suggest the authors try to use terms which indicate the direction of the change: e.g. warmer conditions increase N mineralization...

We assume the reviewer is referring to section 4.1.2. The relevant paragraph has been slightly modified to meet the reviewer's request.

Due to the complicated relationship between climate and N leaching, as described in this section, it is not always clear what is the exact mechanism behind the predicted responses. Therefore, we prefer to formulate the discussion as we did: first mention the overall effect of climate change on N leaching and then discuss the likely reasons behind this response.

**Minor comments:**

- P2, L. 15: "with increasing N input, the capacity of ecosystems to retain N decreases.." that statement surprises me. I can understand that with increasing N input, leaching increases, but the capacity to retain N does not necessarily changes.
  "N input" has been changed to "N availability", which is more accurate.

- P4, L6. Add "be" between can and found.
  Done.

- P6, L. 27, "we"
  Done.

- P6, L.29: remove "to" between by and "the fraction"
  Done.

- Notation of "N leaching : N input ration", I don't; really like that notation as ":" also denotes a ratio. I would suggest to modify to "N leaching/N input" or "N leaching to N input ratio".
  All instances have been changed to "N leaching to N input ratio".

- P8, L. 20: add "is" after variability
  Done.

- Section 3.2.1, L. 18: N deposition and pCO2 increased but not climate, please modify.
  Done.

- P11, L.15, year of citation for Cleveland et al., is missing
  Since we already listed the publication year three lines previous, we deliberately omitted it in this sentence. The phrase has been changed from "the Cleveland et al. N fixation-AET relationship" to "the N fixation-AET relationship of Cleveland et al."

- Figures: not sure that showing the 99% quantile is the best way to go as areas with very large changes do not come out. You could also use a nonlinear color scale.
  The main reason to introduce the cutoff of the color axes in the maps are the extremely high N deposition and N leaching rates in Indonesia, as discussed in sections 3.1.1–3. These high values may not be realistic, and are of less relevance to this study since we are more interested in large-scale patterns. Since using a non-linear scale makes it more difficult to infer the absolute numbers from the graphs, we prefer to keep the color scales as they are.

- Figure 8 and 9: I find it a bit confusing that panel c does not go all the way to 90S. It would be better visually to have the latitudes of the 3 plots match: i.e. if c stops at 60S, then c does not have to be shown over the whole vertical plot and would stop at 60S in panels a and b).
  For all three panels in both figures (and the new figure), the lower limit of latitude is at 60°S.

**Response to comments of reviewer 2**

We thank the reviewer for the helpful comments which have led to an improvement of the paper. Below our reply to the specific comments, set in red font color.

Braakhekke et al. present a model study quantifying the respective contributions of changes in nitrogen (N) deposition, climate and atmospheric CO2 concentration on changes in N leaching from natural ecosystems. They find that increasing N deposition is the major driver behind simulated changes in N leaching, with smaller contribution from climate change and increasing CO2. They further highlight the role of fire in shaping N losses. The conclusions drawn have a sound basis on the the results discussed here. Overall, the manuscript is well written and clearly structured.

However, I find that the discussion of gaseous losses in the manuscript is lacking. Leaching losses are often the major loss, but gaseous losses are not negligible and can regionally dominated total N losses (Houlton et al. 2015). Accordingly, the role of gaseous losses needs to be considered when the ratio of leaching:inputs is discussed.

The author state in the introduction that "N leaching, while sometimes reported in global modelling studies, does generally not receive specific attention [...]". Therefore, due to the lack of evaluation of simulated loss terms, the reliability of global models in respect to the loss terms has to be considered low. This is true for LPJ as previous studies did not evaluate the loss fluxes sufficiently (Smith et al 2014 Warling et al., 2014). Unfortunately, the study by Braakhekke et al. does not improve this situation although data sets exist to evaluate. For example, the simulated gaseous loss fraction can be compared to reconstructions from delta 15 N measurements and models by Houlton et al. (2015) and, more recently Goll et al. (2017). There are regional differences in the dominant loss pathway between this study (Figure 3) and the mentioned studies which should be discussed.

We have added a figure and discussion on comparison of the fraction of N lost by denitrification to two global observation-based datasets: Wang et al. (2017) and Goll et al. (2017) (Figure 10). However, these two datasets differ considerably, demonstrating the current uncertainty regarding denitrification rates, also for observation based estimates.

The role of fire in shaping N loss pathways on global is a novel aspect of this study. The analysis would benefit from information on how simulated fire emissions and the contribution of wildfires to N deposition (forcing) compare to each other. Such information is completely lacking in the manuscript.

We acquired the dataset of N emissions from biomass burning used to derive N deposition. A figure has been added to the supplemental information (Figure S11), and the discussion on the role of fire (section 4.1.4) has been expanded.

In the abstract is stated "Predicted global N leaching from natural lands rose from 13.6 Tg N yr-1 in 1901–1911 to 18.5 Tg N yr-1 in 1997–2006, accounting for land-use changes." (P1L25/26). Did the authors account for land-use change? The information in the manuscript is insufficient to tell to what extent land use (change) and for example associated nitrogen fertilization was accounted for.

Since our study concerns only natural ecosystems, we did not consider N fertilization. However, to determine global total N leaching from natural lands, changes in natural land cover over time (mainly reduction) need to be considered. We did this simply by multiplying the fluxes by the natural landcover fraction for each grid cell. The sentence in the abstract to which the reviewer referred has been modified as to make this more clear: "*Predicted global N leaching from natural lands rose from 13.6 Tg N yr$^{-1}$ in 1901–1911 to 18.5 Tg N yr$^{-1}$ in 1997–2006, accounting for reductions of natural landc*over".

Minor P2L14: reference missing

Reference added.

P4L27: what is the criteria applied to define when the equilibrium state is reached?

During the spinup no checking is done to test how close the model is to equilibrium state. However, for soil organic carbon a root-finding solver is used midway in the spinup, to bring SOC pools very close to equilibrium. Testing has shown that this is sufficient.

P5L12: how are the grass PFTs being more competitive than trees in the model?

The higher competitiveness of grasses is achieved through PFT-specific parametrization, most importantly (cf also Smith et al., 2014):

1) Grasses have a higher uptake capacity per unit root biomass
2) In case insufficient N is available for all PFT (cohorts) N is partitioned among individuals according to a "relative uptake strength", which is higher for grasses.

P11:13: BNF estimates were revised down since Cleveland et al. 1999. Please account for newer estimates here; for example see Vitousek et al. 2013 , Sullivan et al.  2014.

We thank the reviewer for this good advice. The discussion has been updated to include the suggested references.

P14L23: The authors state that N deposition is the dominant factor driving spatial differences in the leaching rate. This needs to shown, as this is not apparent. I rather would suspect differences in the hydrological cycle to dominate spatial leaching patterns.

Our statement was largely based on the results of the factorial experiment (Figures 11 & 12, in the new manuscript). Because of the reviewer's comment we determined spatial covariation based on a moving window approach. This showed that both variation in precipitation and N deposition determine spatial patterns of N leaching, hence our statement was overly reductive. We removed it from the discussion.

P14L26: how is the correction done.  This should be stated in the method sections.

This is described in section 2.3. We added a reference to this section.

Figure 4: the ratios, denitrification:inputs and fire:inputs, would be interesting to see and to better understand the lack of non-linearity in the simulated leaching:input ratio (Figure 6)

The suggested graphs have been added to the supplemental information (Fig S9 & S10).

P14L30: the substantial underestimation of BNF in LPJ should lead to lower leaching rates. This should be discussed.

A sentence has been added to the paragraph.

Reference:  Houlton, Benjamin Z., Alison R. Marklein, and Edith Bai.  "Representation of nitrogen in climate change forecasts." Nature Climate Change 5.5 (2015): 398-401.

Goll, D. S., Winkler, A. J., Raddatz, T., Dong, N., Prentice, I. C., Ciais, P., and Brovkin, V.:  Carbon-nitrogen interactions in idealized simulations with JSBACH (version 3.10), Geosci. Model Dev. Discuss., doi:10.5194/gmd-2016-304, in review, 2017

**Response to comments of reviewer 3**

We thank the reviewer for the helpful comments. Below our reply set in red font color.

This paper presents a modeling study on global nitrogen leaching from natural ecosystems with an ecosystem model LPJ-GUESS. Overall, the paper is well-written and the results are informative. My major concern is on the dependence of results on model representation of nitrogen and carbon cycling process and the inherent model assumptions. Please find my specific comments below.

1. Authors conclude that atmospheric N deposition is the major driver behind nitrogen leaching globally. This is not surprising as atmospheric nitrogen deposition is the dominant N input and linearly linked to soil nitrogen storage in the model. I would suggest add more details on the mathematical formulations in representing atmospheric nitrogen deposition and nitrogen mineralization in the model. Are the results sensitive to the specific formulation of atmospheric nitrogen deposition and mineralization? Discussion on the nitrogen deposition dataset should also be added.

Nitrogen deposition was not predicted in this study, but taken from the dataset of the Atmospheric Chemistry and Climate Model Intercomparison Project (ACCMIP; Lamarque et al., 2013). This data comprises results from an ensemble of simulations with 11 complex atmospheric chemistry models. Therefore, providing details on the mathematical formulation of N deposition is not feasible. For this we refer to the Lamarque et al., (2013) and references therein. The results of the ACCMIP simulations have been thoroughly and favourably evaluated against observations by Lamarque et al. (2013), and can be assumed to represent the best estimate of global N deposition currently available. A sentence stating this has been added to section 2.2.1.

Regarding N mineralization, indeed the results are likely to be sensitive to the mathematical formulations in soil N cycling module. The soil N cycling module in LPJ-GUESS is largely based on the CENTURY model which has been applied many in many studies, and is described in detail in Smith et al. (2014) and Parton et al. (1993). A detailed discussion on the effect of different formulations of mineralization is outside the scope of this paper, also because we deem other aspects of the model more relevant for uncertainty of predicted N leaching (discussed in section 4.4). A thorough comparison of different N cycling models (including LPJ-GUESS) has been published by Zaehle et al. (2014), including discussions on the formulation of N mineralization.

2. Table 1 lists the numerical experiments conducted in this study. I would suggest add some statements on the purpose of these experiment designs in the methodology section 2.3. For example, which combination of experiments is used to disentangle the effects of a specific environmental driver (e.g. N deposition)?

Section 2.3 has been expanded with several sentences.

3. Increase in nitrogen deposition may potentially lead to increased plant carbon uptake and plant productivity, which would feedback to the nitrogen budget and affect leaching process. Is the impact of carbon–nitrogen interactions on N leaching process considered in this study? This aspect of carbon–nitrogen dynamics on N leaching process should at least be discussed. Please also add a table specifying

carbon–nitrogen ratios for all natural plant types considered. It would be interesting to examine/discuss how plant growth regulate the simulated N leaching for different PFTs in the model.

LPJ-GUESS is a dynamic global ecosystem model that simulates fully coupled C and N cycling in vegetation and soil. Hence, C-N interactions are considered and indeed an important aspect of this study (cf the effects of N deposition and $CO_2$ on GPP, shown in supplemental figure S19).

Leaf C-N ratios are calculated prognostically based on canopy level photosynthesis and N availability (as described in supplement text S1), while C-N ratios of other pools are fixed. We added a figure of the PFT-mean leaf C:N ratios to the supplemental information (Figure S12)

4. The effects of fire and gaseous loss on N leaching is analyzed in the results section. But the descriptions on the representation of fire in the model is missing. In addition, it seems that the proposed numerical experiments in Table 1 doesn't consider fire?

Description of the fire module is given in supplemental text S1. This section has been expanded.

5.  In section 2.2.1, the CRU monthly climate is interpolated to daily values as inputs for the model. More details on this temporal disaggregation are required. Discussions are also needed as the simulated sensitivity of N leaching to precipitation may depend on the daily sequence of precipitation and intensity.

Additional information on the interpolation of the climate data has been added to section 2.2.1. A sentence stating the relevance of rainfall distribution in time for N leaching has been added to section 4.1.2.

6. I would suggest clarify which specific aspect of N leaching is the focus of this study, the mean value or its temporal variation?

A sentence has been added to the introduction: "*Specifically, we focus on temporal changes during the last century in relation to change of environmental drivers, as well as spatial patterns of contemporary N leaching rates.*"

7. How is N status quantified?

This is done based on the N limitation factor, expressing reduction of photosynthesis due to N limitation. This is described in sections 2.1.2 and 2.3.

8.  In section 3.2.1, the statement "N deposition, climate and atmospheric $CO_2$ all increased during the 20th century" is confusing as "climate" is a broad concept.

The sentence has been modified.

9. I would suggest using the percentage change (%) as the unit in Figure 10

We assume the reviewer suggests to plot the relative changes as (sim-control)/control. We attempted this but it does not result in a readable graph, because regions where N leaching is very low in the control the relative change takes very high values, often infinity or NaN.

10. The name of the model used in this study can be added in the title.

We prefer to leave the title as it is. The name of the model is mentioned in the abstract.

[revised manuscript text omitted]

M. C. Braakhekke[1,2], K.T. Rebel[1], S.C. Dekker[1], B. Smith[2], A.H.W. Beusen[2,4], and M.J. Wassen[1]

[1]Copernicus Institute of Sustainable Development, Faculty of Geosciences, Utrecht University, Heidelberglaan 2, 3584 CS, Utrecht, the Netherlands
[2]PBL Netherlands Environmental Assessment Agency, Postbus 30314, 2500 GH, The Hague, the Netherlands
[3]Department of Physical Geography and Ecosystem Science, Lund University, 22362, Lund, Sweden.
[4]Department of Earth Sciences, Geochemistry, Faculty of Geosciences, Utrecht University, P.O. Box 80021, 3508 TA, Utrecht, the Netherlands

**Contents of this file**

**Text S1. Description of LPJ-GUESS**

*General description*

LPJ-GUESS (Lund-Potsdam-Jena General Ecosystem Simulator; Smith et al., 2001) simulates vegetation dynamics and biogeochemical fluxes of C and N in terrestrial ecosystems and employs generalized biome- or global-scale parameterizations of component ecosystem processes, allowing it to be employed without recalibration globally or for any large region. It is forced by climate variables, $CO_2$ concentration and N deposition and runs with a daily time step, except for C allocation, vegetation dynamics, and disturbances, which are resolved annually. Our simulations focused on natural vegetation, i.e. croplands were not considered. Eleven plant functional types (PFTs) were included, representing vegetation in temperate, tropical, boreal, and grassland biomes. The model predicts the occurrence of each PFT based on bioclimatic limits and competition with other PFTs for light and soil resources. Contrary to most global ecosystem models, LPJ-GUESS explicitly represents the age distribution of woody PFTs. The model simulates trees of different cohorts (age classes) which are each represented by an average individual for each age class of each of a number of co-occurring PFTs. Mortality and establishment of the individuals are implemented in a stochastic fashion, as are fire and other disturbances (see below). Sub-grid variability resulting from landscape heterogeneity and differences in disturbance history are accounted for by simulating a predefined number of replicate "patches" (area 0.1 ha) per grid cell. The conditions for all patches within a grid cell are identical but differences arise from the stochastic calculations.

Within each patch LPJ-GUESS simulates fluxes of C, water and N, in vegetation and soil based on descriptions of various processes, including photosynthesis, plant C allocation, autotrophic respiration, evapotranspiration, percolation, lateral runoff, and soil carbon cycling. Soil carbon cycling is simulated using a scheme based on the CENTURY model, as described by Smith et al. (2014) . Soil hydrology is represented using two soil layers of 0.5 and 1 m. Downward percolation is simulated with a leaky bucket scheme (Gerten et al., 2004). Available water capacity (AWC) and a hydraulic conductivity are derived from sand, silt and clay fraction, as described in Olin et al. (2015). Water in excess of the AWC in the first and second soil layer is exported as surface runoff and interflow, respectively.

Fire is modelled stochastically according to the scheme described in Thonicke et al. (2001) . Fire can occur when a fuel (litter) load of 200 g m$^{-2}$ or higher is present. When this is the case, the probability of a fire  occurring on a given day is a non-linear function of the moisture content of the upper soil layer, serving as proxy for the litter layer. The actual occurrence of a fire is determined using a random number generator. If a fire occurs, a PFT dependent fraction of the biomass (50–90 %) is lost from the ecosystem. Additional disturbances, killing all vegetation, are modelled with a fixed expected return time of 100 years.

The simulation is initialized with a 500-year spin-up to accumulate vegetation and soil C pools in equilibrium with the initial forcing. During this phase, the model is forced by a trend-free time series (here 10 years; see below) of annually-varying inputs.

*N cycling module*

In LPJ-GUESS ecosystem N is present in vegetation biomass and in the soil in mineral and organic form. In the model version employed for our study, mineral soil N is represented by a single pool; i.e. different N species such as ammonium and nitrate, and transformation between these are not distinguished. Atmospheric N deposition is added to the soil mineral N pool. Biological N fixation (BNF) is calculated as a linear function of evapotranspiration (Cleveland et al., 1999) and added to the mineral N pool, up to a maximum pool capacity of 2 g N m$^{-2}$. Root uptake transfers N from the soil mineral N pool to vegetation on a daily time step. Plant N demand is driven by optimal leaf N content required for photosynthesis, computed based on the carboxylation capacity of Rubisco ($V_{max}$) that maximizes canopy-level net photosynthesis (Haxeltine and Prentice, 1996), given current atmospheric $CO_2$ concentration, temperature, soil water, and leaf area index (LAI). In addition to leaves plants require N for sapwood, heartwood, and roots. Plants also maintain an N store pool with a maximum size defined by biomass, leaf C:N ratio, and PFT type. Following Meyerholt and Zaehle (2015), the C:N ratios of non-leaf pools are fixed, which represents a modification to the model version described in Smith et al. (2014), where C:N ratios of non-leaf tissue were scaled based on leaf C:N. Plants take up N from the mineral soil pool and the N store in order to maintain optimal leaf N. If insufficient N is available the plant experiences N stress and $V_{max}$ is reduced. To this end the model calculates an "N limitation factor" equal to the ratio of the true $V_{max}$ and the $V_{max}$ in absence of N limitation (both without water limitation). Additionally, different PFT cohorts compete for N, where the competitive strength of an individual is determined by its root biomass, the combined C:N ratio of roots and leaves, and growth form, with grass PFTs being more competitive than tree PFTs.

N stored in vegetation is returned to the soil in organic form in conjunction with biomass turnover due to senescence, mortality, and disturbance. Retranslocation transfers 50 % of leaf and root N to the N store pool, up to a maximum capacity, prior to turnover. Litter and soil organic matter (SOM) dynamics follow the CENTURY model (Parton et al., 1993), which includes nine litter and SOM pools, and two microbial pools. The C:N ratio of the slowest SOM pool (passive) is fixed, while the other pools vary depending on litter N content and soil available N. During decomposition N is transferred to or from the mineral N in order to maintain the C:N ratio of the pools. If insufficient N is available decomposition rates are reduced. Additionally 1 % of the daily N mineralization is lost, representing gaseous N loss during nitrification and denitrification. Organic N leaching occurs as a fraction of the soil microbial N pool, determined by the percolation rate and the sand fraction. Mineral N leaching is calculated as a fraction of the mineral N pool equal to the relative water loss by percolation and interflow. Surface runoff does not cause N loss. Finally, fire events cause loss of vegetation N which is assumed to be emitted in gaseous form.

[Figure]

**Figure S1.** Change of model drivers during the simulation period. (a) global total atmospheric N deposition; (b) global mean temperature; (c) global mean precipitation rate; (d) global mean atmospheric $CO_2$ concentration.

[Figure]

**Figure S2.** Mean temperature difference between 1997–2006 and 1901–1910 (°C).

[Figure]

**Figure S3.** Mean precipitation difference between 1997–2006 and 1901–1910 (mm yr$^{-1}$).

[Figure]

**Figure S4.** Biome distribution for the true historical simulation (+Ndep +clim +CO$_2$). Biome classes are derived from leaf area index of the PFTs averaged over the period 1997–2006 (Smith et al., 2014).

[Figure]

**Figure S5.** Biological N fixation (kg-N ha$^{-1}$ yr$^{-1}$) for the true historical simulation (+Ndep +clim +CO$_2$) averaged over the period 1997–2006.

[Figure]

**Figure S6.** Organic N leaching (kg N ha$^{-1}$ yr$^{-1}$) for the true historical simulation (+Ndep +clim +CO$_2$) averaged over the period 1997–2006.

[Figure]

**Figure S7.** N leaching relative to N input vs N status for the true historical simulation (+Ndep +clim +CO₂) averaged over the period 1997–2006. Colors indicate the combined rate of interflow and percolation from the root zone (mm yr⁻¹; upper limit cut off to improve readability)

[Figure]

**Figure S8.** N status vs total N input (fixation + deposition) for the true historical simulation (+Ndep +clim +CO₂) averaged over the period 1997–2006.

[Figure]

**(a) Absolute (kg ha⁻¹ yr⁻¹)**

[Figure]

[Figure]

**Figure S9.** N loss due to  denitrification (kg ha⁻¹ yr⁻¹) for the true historical simulation (+Ndep +clim +CO$_2$) averaged over the period 1997–2006. (a) Absolute; (b) Relative to total N input (deposition + fixation)

[Figure]

[Figure]

**Figure S10.** N loss due to fire (kg N ha$^{-1}$ yr$^{-1}$) for the true historical simulation (+Ndep +clim +CO$_2$) averaged over the period 1997–2006. (a) Absolute; (b) Relative to total N input (deposition + fixation)

[Figure]

**Figure S11.** Fire N emissions (kg N ha$^{-1}$ yr$^{-1}$) input in the atmospheric chemistry models that were used to derive the ACCMIP dataset of atmospheric N deposition. Note that the axis scale has been cutoff at 12 kg N ha$^{-1}$ yr$^{-1}$ for comparability with Figure S10a.

[Figure]

**Figure S12.** Predicted leaf C to N ratio of the plant functional types in LPJ-GUESS. The bars show means over the grid cells where the respective PFTs are dominant (have highest leaf area index). Error bars indicate 1 standard deviation.

[Figure]

**Figure S13.** Soil organic carbon storage (kg C m$^{-2}$) for the true historical simulation (+Ndep +clim +CO$_2$) averaged over the period 1997–2006.

[Figure]

**Figure S1411.** N leaching relative to total N input vs N status averaged over the period 1997–2006 for a simulation run with true drivers (+Ndep +clim +CO$_2$) but fire and other disturbances switched off. TrRF: Tropical rainforest, TrDF: Tropical deciduous forest, TrSF: Tropical seasonal forest, BEF: Boreal evergreen forest/woodland, BDF: Boreal deciduous forest/woodland, TeBEF: Temperate broadleaved evergreen forest, TeDF: Temperate deciduous forest, TeBMF: Temperate/boreal mixed forest, TeMF: Temperate mixed forest, XWS: Xeric woodland/shrubland, MS: Moist savannah, DS: Dry savannah, AT: Arctic/alpine tundra, TG: Tall grassland, AS: Arid shrubland/steppe, DG: Dry grassland, Des: Desert.

[Figure]

**Figure S15.** Comparison of N fluxes with results of Beusen et al., (2016). (a) Total global N deposition, including non-natural lands. (b) Total global N deposition from natural lands, corrected for changes in natural landcover based on the dataset used by Beusen et al. (2016).

[Figure]

**Figure S16.** N leaching difference with control simulation (-Ndep -clim -CO$_2$) for the eight simulations averaged over the period 1997–2006.

[Figure]

**Figure S17.** Mineral N leaching difference with the control simulation (-Ndep -clim -CO₂) for the two-factor simulations. For readability, the color axis has been cut off at approximately the 1% and 99% quantile.

[Figure]

**Figure S1815.** Global total GPP vs time for the eight simulations. For readability, the time series have been smoothed with a 5 year moving window

[Figure]

**Figure S1916.** GPP difference (kg-C m$^{-2}$ yr$^{-1}$) with control simulation (-Ndep -clim -CO$_2$) for the other simulations averaged over the period 1997–2006.

[Figure]

**Figure S20.** N leaching vs N deposition in European temperate deciduous forests for the true historical simulation (+Ndep +clim +CO₂) (averaged over the period 1997–2006) and Level II sites of the UN-ECE/EC Intensive Monitoring Programme (Dise et al., 2009). Colors indicate ecosystem N status.

[Figure]

**Figure S21.** Mineral N leaching vs runoff for North and South America. The linear fit on log-log scale is compared to a fit for dissolved inorganic nitrogen (DIN) losses vs runoff published by Lewis et al. (1999).

[Figure]

**Figure S22.** Global total net N mineralization vs time for the eight simulations. For readability, the time series have been smoothed with a 5 year moving window.

| | TrRF | TrDF | TrSF | BEF | BDF | TeBEF | TeDF | TeBMF | TeMF | XWS | MS | DS | AT | TG | AS | DG | Des |
|---|---|---|---|---|---|---|---|---|---|---|---|---|---|---|---|---|---|
| **TrRF** | - | 0.655 | 0.000 | 0.000 | 0.000 | 0.063 | 0.000 | 0.000 | 0.728 | 0.000 | 0.000 | 0.000 | 0.000 | 0.000 | 0.000 | 0.000 | 0.000 |
| **TrDF** | 0.655 | - | 0.000 | 0.000 | 0.000 | 0.040 | 0.000 | 0.000 | 0.557 | 0.000 | 0.000 | 0.000 | 0.000 | 0.000 | 0.000 | 0.000 | 0.000 |
| **TrSF** | 0.000 | 0.000 | - | 0.000 | 0.000 | 0.001 | 0.000 | 0.000 | 0.142 | 0.000 | 0.000 | 0.000 | 0.000 | 0.000 | 0.000 | 0.000 | 0.000 |
| **BEF** | 0.000 | 0.000 | 0.000 | - | 0.000 | 0.000 | 0.000 | 0.000 | 0.000 | 0.000 | 0.000 | 0.000 | 0.000 | 0.000 | 0.000 | 0.000 | 0.000 |
| **BDF** | 0.000 | 0.000 | 0.000 | 0.000 | - | 0.000 | 0.000 | 0.000 | 0.000 | 0.000 | 0.000 | 0.000 | 0.000 | 0.000 | 0.000 | 0.000 | 0.000 |
| **TeBEF** | 0.063 | 0.040 | 0.001 | 0.000 | 0.000 | - | 0.000 | 0.000 | 0.384 | 0.000 | 0.000 | 0.000 | 0.000 | 0.000 | 0.000 | 0.000 | 0.000 |
| **TeDF** | 0.000 | 0.000 | 0.000 | 0.000 | 0.000 | 0.000 | - | 0.000 | 0.047 | 0.000 | 0.000 | 0.000 | 0.000 | 0.000 | 0.000 | 0.000 | 0.000 |
| **TeBMF** | 0.000 | 0.000 | 0.000 | 0.000 | 0.000 | 0.000 | 0.000 | - | 0.043 | 0.000 | 0.000 | 0.000 | 0.000 | 0.000 | 0.000 | 0.000 | 0.000 |
| **TeMF** | 0.728 | 0.557 | 0.142 | 0.000 | 0.000 | 0.384 | 0.047 | 0.043 | - | 0.000 | 0.001 | 0.000 | 0.000 | 0.000 | 0.000 | 0.000 | 0.000 |
| **XWS** | 0.000 | 0.000 | 0.000 | 0.000 | 0.000 | 0.000 | 0.000 | 0.000 | 0.000 | - | 0.000 | 0.000 | 0.000 | 0.001 | 0.000 | 0.000 | 0.000 |
| **MS** | 0.000 | 0.000 | 0.000 | 0.000 | 0.000 | 0.000 | 0.000 | 0.000 | 0.001 | 0.000 | - | 0.000 | 0.000 | 0.000 | 0.000 | 0.000 | 0.000 |
| **DS** | 0.000 | 0.000 | 0.000 | 0.000 | 0.000 | 0.000 | 0.000 | 0.000 | 0.000 | 0.000 | 0.000 | - | 0.000 | 0.000 | 0.000 | 0.000 | 0.000 |
| **AT** | 0.000 | 0.000 | 0.000 | 0.000 | 0.000 | 0.000 | 0.000 | 0.000 | 0.000 | 0.000 | 0.000 | 0.000 | - | 0.000 | 0.000 | 0.000 | 0.000 |
| **TG** | 0.000 | 0.000 | 0.000 | 0.000 | 0.000 | 0.000 | 0.000 | 0.000 | 0.000 | 0.001 | 0.000 | 0.000 | 0.000 | - | 0.000 | 0.000 | 0.000 |
| **AS** | 0.000 | 0.000 | 0.000 | 0.000 | 0.000 | 0.000 | 0.000 | 0.000 | 0.000 | 0.000 | 0.000 | 0.000 | 0.000 | 0.000 | - | 0.000 | 0.000 |
| **DG** | 0.000 | 0.000 | 0.000 | 0.000 | 0.000 | 0.000 | 0.000 | 0.000 | 0.000 | 0.000 | 0.000 | 0.000 | 0.000 | 0.000 | 0.000 | - | 0.000 |
| **Des** | 0.000 | 0.000 | 0.000 | 0.000 | 0.000 | 0.000 | 0.000 | 0.000 | 0.000 | 0.000 | 0.000 | 0.000 | 0.000 | 0.000 | 0.000 | 0.000 | - |

**Table S1.** Test statistic of Welch's t-test for determining differences between biomes in total ecosystem N input (mean 1997–2006) between biomes (c.f. Fig. 2, main text). Welch's t-test is used for comparing populations with different variances with different and samples sizes. Significant differences ($\alpha \leq 0.05$) are printed in red. TrRF: Tropical rainforest, TrDF: Tropical deciduous forest, TrSF: Tropical seasonal forest, BEF: Boreal evergreen forest/woodland, BDF: Boreal deciduous forest/woodland, TeBEF: Temperate broadleaved evergreen forest, TeDF: Temperate deciduous forest, TeBMF: Temperate/boreal mixed forest, TeMF: Temperate mixed forest, XWS: Xeric woodland/shrubland, MS: Moist savannah, DS: Dry savannah, AT: Arctic/alpine tundra, TG: Tall grassland, AS: Arid shrubland/steppe, DG: Dry grassland, Des: Desert.

|      | TrRF | TrDF | TrSF | BEF | BDF | TeBEF | TeDF | TeBMF | TeMF | XWS | MS | DS | AT | TG | AS | DG | Des |
|------|------|------|------|-----|-----|-------|------|-------|------|-----|-----|-----|-----|-----|-----|-----|-----|
| **TrRF** | - | 0.830 | 0.005 | 0.000 | 0.000 | 0.027 | 0.000 | 0.000 | 0.018 | 0.000 | 0.002 | 0.000 | 0.000 | 0.000 | 0.000 | 0.000 | 0.000 |
| **TrDF** | 0.830 | - | 0.027 | 0.000 | 0.000 | 0.082 | 0.001 | 0.000 | 0.000 | 0.000 | 0.019 | 0.000 | 0.000 | 0.000 | 0.000 | 0.000 | 0.000 |
| **TrSF** | 0.005 | 0.027 | - | 0.000 | 0.000 | 0.000 | 0.192 | 0.000 | 0.000 | 0.000 | 0.442 | 0.000 | 0.000 | 0.000 | 0.000 | 0.000 | 0.000 |
| **BEF** | 0.000 | 0.000 | 0.000 | - | 0.000 | 0.000 | 0.000 | 0.000 | 0.000 | 0.000 | 0.000 | 0.000 | 0.000 | 0.000 | 0.000 | 0.000 | 0.000 |
| **BDF** | 0.000 | 0.000 | 0.000 | 0.000 | - | 0.000 | 0.000 | 0.000 | 0.000 | 0.000 | 0.000 | 0.000 | 0.000 | 0.000 | 0.000 | 0.000 | 0.000 |
| **TeBEF** | 0.027 | 0.082 | 0.000 | 0.000 | 0.000 | - | 0.000 | 0.000 | 0.004 | 0.000 | 0.000 | 0.000 | 0.000 | 0.000 | 0.000 | 0.000 | 0.000 |
| **TeDF** | 0.000 | 0.001 | 0.192 | 0.000 | 0.000 | 0.000 | - | 0.000 | 0.000 | 0.000 | 0.011 | 0.000 | 0.000 | 0.000 | 0.000 | 0.000 | 0.000 |
| **TeBMF** | 0.000 | 0.000 | 0.000 | 0.000 | 0.000 | 0.000 | 0.000 | - | 0.476 | 0.000 | 0.000 | 0.000 | 0.000 | 0.000 | 0.000 | 0.000 | 0.000 |
| **TeMF** | 0.018 | 0.000 | 0.000 | 0.000 | 0.000 | 0.004 | 0.000 | 0.476 | - | 0.000 | 0.000 | 0.000 | 0.000 | 0.000 | 0.000 | 0.000 | 0.000 |
| **XWS** | 0.000 | 0.000 | 0.000 | 0.000 | 0.000 | 0.000 | 0.000 | 0.000 | 0.000 | - | 0.000 | 0.000 | 0.000 | 0.000 | 0.000 | 0.000 | 0.000 |
| **MS** | 0.002 | 0.019 | 0.442 | 0.000 | 0.000 | 0.000 | 0.011 | 0.000 | 0.000 | 0.000 | - | 0.000 | 0.000 | 0.000 | 0.000 | 0.000 | 0.000 |
| **DS** | 0.000 | 0.000 | 0.000 | 0.000 | 0.000 | 0.000 | 0.000 | 0.000 | 0.000 | 0.000 | 0.000 | - | 0.000 | 0.000 | 0.000 | 0.000 | 0.000 |
| **AT** | 0.000 | 0.000 | 0.000 | 0.000 | 0.000 | 0.000 | 0.000 | 0.000 | 0.000 | 0.000 | 0.000 | 0.000 | - | 0.000 | 0.000 | 0.000 | 0.000 |
| **TG** | 0.000 | 0.000 | 0.000 | 0.000 | 0.000 | 0.000 | 0.000 | 0.000 | 0.000 | 0.000 | 0.000 | 0.000 | 0.000 | - | 0.000 | 0.000 | 0.000 |
| **AS** | 0.000 | 0.000 | 0.000 | 0.000 | 0.000 | 0.000 | 0.000 | 0.000 | 0.000 | 0.000 | 0.000 | 0.000 | 0.000 | 0.000 | - | 0.000 | 0.000 |
| **DG** | 0.000 | 0.000 | 0.000 | 0.000 | 0.000 | 0.000 | 0.000 | 0.000 | 0.000 | 0.000 | 0.000 | 0.000 | 0.000 | 0.000 | 0.000 | - | 0.000 |
| **Des** | 0.000 | 0.000 | 0.000 | 0.000 | 0.000 | 0.000 | 0.000 | 0.000 | 0.000 | 0.000 | 0.000 | 0.000 | 0.000 | 0.000 | 0.000 | 0.000 | - |

**Table S2.** Test statistic of Welch's t-test for determining differences between biomes in total ecosystem N loss (mean 1997–2006) between biomes (c.f. Fig. 2, main text). Welch's t-test is used for comparing populations with different variances with different and samples sizes. Significant differences (α ≤ 0.05) are printed in red. TrRF: Tropical rainforest, TrDF: Tropical deciduous forest, TrSF: Tropical seasonal forest, BEF: Boreal evergreen forest/woodland, BDF: Boreal deciduous forest/woodland, TeBEF: Temperate broadleaved evergreen forest, TeDF: Temperate deciduous forest, TeBMF: Temperate/boreal mixed forest, TeMF: Temperate mixed forest, XWS: Xeric woodland/shrubland, MS: Moist savannah, DS: Dry savannah, AT: Arctic/alpine tundra, TG: Tall grassland, AS: Arid shrubland/steppe, DG: Dry grassland, Des: Desert.

| | TrRF | TrDF | TrSF | BEF | BDF | TeBEF | TeDF | TeBMF | TeMF | XWS | MS | DS | AT | TG | AS | DG | Des |
|---|---|---|---|---|---|---|---|---|---|---|---|---|---|---|---|---|---|
| **TrRF** | - | 0.231 | 0.106 | 0.000 | 0.374 | 0.000 | 0.000 | 0.383 | 0.000 | 0.950 | 0.000 | 0.000 | 0.000 | 0.000 | 0.000 | 0.000 | 0.000 |
| **TrDF** | 0.231 | - | 0.034 | 0.889 | 0.012 | 0.000 | 0.000 | 0.127 | 0.000 | 0.331 | 0.000 | 0.000 | 0.000 | 0.000 | 0.000 | 0.000 | 0.000 |
| **TrSF** | 0.106 | 0.034 | - | 0.000 | 0.329 | 0.001 | 0.000 | 0.528 | 0.000 | 0.378 | 0.000 | 0.000 | 0.000 | 0.000 | 0.000 | 0.000 | 0.000 |
| **BEF** | 0.000 | 0.889 | 0.000 | - | 0.000 | 0.000 | 0.000 | 0.000 | 0.000 | 0.130 | 0.000 | 0.000 | 0.000 | 0.000 | 0.000 | 0.000 | 0.000 |
| **BDF** | 0.374 | 0.012 | 0.329 | 0.000 | - | 0.000 | 0.000 | 0.932 | 0.000 | 0.543 | 0.000 | 0.000 | 0.000 | 0.000 | 0.000 | 0.000 | 0.000 |
| **TeBEF** | 0.000 | 0.000 | 0.001 | 0.000 | 0.000 | - | 0.000 | 0.000 | 0.006 | 0.002 | 0.000 | 0.000 | 0.000 | 0.000 | 0.000 | 0.000 | 0.000 |
| **TeDF** | 0.000 | 0.000 | 0.000 | 0.000 | 0.000 | 0.000 | - | 0.000 | 0.000 | 0.000 | 0.000 | 0.000 | 0.000 | 0.000 | 0.000 | 0.000 | 0.000 |
| **TeBMF** | 0.383 | 0.127 | 0.528 | 0.000 | 0.932 | 0.000 | 0.000 | - | 0.001 | 0.693 | 0.000 | 0.000 | 0.000 | 0.000 | 0.000 | 0.000 | 0.000 |
| **TeMF** | 0.000 | 0.000 | 0.000 | 0.000 | 0.000 | 0.006 | 0.000 | 0.001 | - | 0.000 | 0.026 | 0.006 | 0.099 | 0.025 | 0.059 | 0.001 | 0.951 |
| **XWS** | 0.950 | 0.331 | 0.378 | 0.130 | 0.543 | 0.002 | 0.000 | 0.693 | 0.000 | - | 0.000 | 0.000 | 0.000 | 0.000 | 0.000 | 0.000 | 0.000 |
| **MS** | 0.000 | 0.000 | 0.000 | 0.000 | 0.000 | 0.000 | 0.000 | 0.000 | 0.026 | 0.000 | - | 0.058 | 0.000 | 0.396 | 0.190 | 0.001 | 0.000 |
| **DS** | 0.000 | 0.000 | 0.000 | 0.000 | 0.000 | 0.000 | 0.000 | 0.000 | 0.006 | 0.000 | 0.058 | - | 0.000 | 0.009 | 0.010 | 0.000 | 0.000 |
| **AT** | 0.000 | 0.000 | 0.000 | 0.000 | 0.000 | 0.000 | 0.000 | 0.000 | 0.099 | 0.000 | 0.000 | 0.000 | - | 0.000 | 0.002 | 0.000 | 0.000 |
| **TG** | 0.000 | 0.000 | 0.000 | 0.000 | 0.000 | 0.000 | 0.000 | 0.000 | 0.025 | 0.000 | 0.396 | 0.009 | 0.000 | - | 0.404 | 0.114 | 0.000 |
| **AS** | 0.000 | 0.000 | 0.000 | 0.000 | 0.000 | 0.000 | 0.000 | 0.000 | 0.059 | 0.000 | 0.190 | 0.010 | 0.002 | 0.404 | - | 0.615 | 0.000 |
| **DG** | 0.000 | 0.000 | 0.000 | 0.000 | 0.000 | 0.000 | 0.000 | 0.000 | 0.001 | 0.000 | 0.001 | 0.000 | 0.000 | 0.114 | 0.615 | - | 0.000 |
| **Des** | 0.000 | 0.000 | 0.000 | 0.000 | 0.000 | 0.000 | 0.000 | 0.000 | 0.951 | 0.000 | 0.000 | 0.000 | 0.000 | 0.000 | 0.000 | 0.000 | - |

**Table S3.** Test statistic of Welch's t-test for determining differences between biomes in N net ecosystem exchange (mean 1997–2006) between biomes (c.f. Fig. 2, main text). Welch's t-test is used for comparing populations with different variances with different and samples sizes. Significant differences ($\alpha \leq 0.05$) are printed in red. TrRF: Tropical rainforest, TrDF: Tropical deciduous forest, TrSF: Tropical seasonal forest, BEF: Boreal evergreen forest/woodland, BDF: Boreal deciduous forest/woodland, TeBEF: Temperate broadleaved evergreen forest, TeDF: Temperate deciduous forest, TeBMF: Temperate/boreal mixed forest, TeMF: Temperate mixed forest, XWS: Xeric woodland/shrubland, MS: Moist savannah, DS: Dry savannah, AT: Arctic/alpine tundra, TG: Tall grassland, AS: Arid shrubland/steppe, DG: Dry grassland, Des: Desert.